# Multiclass Alignment of Confidences and Softened Target Occurrences for Train-time Calibration

## Abstract

In spite of delivering remarkable predictive accuracy across many domains, including computer vision and medical imaging, Deep Neural Networks (DNNs) are susceptible to making overconfident predictions. This could potentially limit their utilization and adoption in many real-world applications, especially involving security-sensitive decision making. Among existing approaches to model calibration, post-hoc based techniques are simple and effective, however, they require a separate hold-out data. Lately, train-time calibration has emerged as an alternate paradigm, in which the recent methods have shown state-of-the-art calibration results. Inspired by the train-time calibration direction, in this paper, we propose a novel train-time calibration method at the core of which is an auxiliary loss formulation, namely multiclass alignment of confidences with the gradually softened ground truth occurrences (MACSO). It is developed on the intuition that, for a class, the gradually softened ground truth occurrences distribution is a suitable non-zero entropy signal whose better alignment with the predicted confidences distribution is positively correlated with reducing the model calibration error. In our train-time approach, besides simply aligning the two distributions, e.g., via their means or KL divergence, we propose to quantify the linear correlation between the two distributions which preserves the relations among them, thereby further improving the calibration performance. Extensive results on several challenging datasets, featuring in and out-of-domain scenarios, class imbalanced problem, and a medical image classification task, validate the efficacy of our method against state-of-the-art train-time calibration methods.

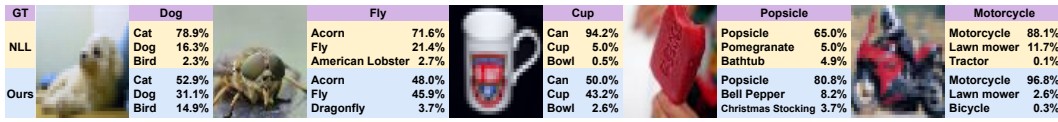

Figure 1: Compared to the negative log-likelihood (NLL) loss, NLL+MACSO (ours) has higher confidences for correct predictions (Popsicle/Motorcycle) and lower confidences for incorrect predictions (Dog/Fly/Cup).

## 1 Introduction

Despite enabling remarkable predictive accuracy for several computer vision tasks, e.g., image classification (Simonyan & Zisserman, 2014; He et al., 2016; Krizhevsky et al., 2009; Dosovitskiy et al., 2020), deep neural networks (DNNs) have the tendency to provide overconfident predictions (Guo et al., 2017; Ovadia et al., 2019; Mukhoti et al., 2020). This renders them poorly calibrated owing to the discrepancy between the predicted confidence of a class and its actual likelihood of occurrence. Due to the increasing deployment of DNN-based models, it is of paramount importance that the model predictions should be well-calibrated for enhancing decision-making and risk assessment. Beyond fostering an overall trust in AI systems, well-calibrated models would enable broader adoption and more effective utilization across diverse domains, including healthcare (Dusenberry et al., 2020; Sharma et al., 2017), autonomous vehicles (Grigorescu et al., 2020), finance (Bao et al., 2017; Dixon et al., 2017), and climate science (Ham et al., 2019; Rasp et al., 2018).

In this paper, we chose the train-time calibration route. We propose a new auxiliary loss formulation, **M**ulticlass **A**lignment of **C**onfidences and **S**oftened ground truth **O**ccurrences (MACSO). It is based on the intuition that, for a class, the gradually softened ground truth occurrences distribution provides a suitable non-zero entropy signal, and it is positively correlated with reducing the model calibration error when the *softened ground truth occurrences distribution* for a class and *predicted confidence distribution* for the same class are more aligned. Besides simply aligning the two distributions using their means or KL divergence, which improves model calibration as shown in our experiments, we propose to measure the linear correlation between them, which preserve the relative order of preferences between the two, and as a result, allows further improving the model calibration. In summary, we make the following contributions:

1. We propose a train-time calibration method with a novel auxiliary loss function, dubbed MACSO, that performs multiclass alignment of confidences and the corresponding gradually softened target occurrences. MACSO can be used with any task-specific loss functions (e.g., NLL), as it is differentiable and operates over minibatches (Fig. 1).
2. Beyond simply aligning the two distributions via their means or KL divergence, we propose to measure the correlation between the two distributions which preserves the relations in each distribution. Empirical evidence shows that this measure further improves calibration.
3. We show that MACSO has desirable theoretical properties. First, in the infinite data limit, MACSO encourages the model to predict the true class probabilities conditional on the features (proper scoring rule). Second, the gradients of the MACSO loss provide an implicit regularization effect during training. These properties help explain MACSO's performance.
4. Extensive experiments have been performed on three in-domain scenarios, CIFAR10/100 (Krizhevsky et al., 2009), and Tiny-ImangeNet (Deng et al., 2009), a class-imbalanced scenario SVHN (Netzer et al., 2011), and three out-of-domain scenarios, CIFAR10/100-C (OOD) (Hendrycks & Gimpel, 2016) and Tiny-ImageNet-C (OOD) (Hendrycks & Gimpel, 2016). Results show that our approach consistently provides improved calibration for both in-domain and out-of-domain predictions compared to the existing state-of-the-art train-time methods. We also demonstrate the effectiveness and applicability of our method on the medical image classification task (Mendeley V2 dataset (Kermany et al., 2018)).

## 2 RELATED WORK

**Post-hoc calibration approaches:** Among the simplest post-hoc calibration approaches, Temperature scaling (TS) is a well-known method that learns a single temperature parameter using a hold-out validation set and then employs it to re-scale the logits values of a trained network (Guo et al., 2017). As a result, the entropy of the logit distribution is increased which eventually helps in improving the model calibration. An obvious limitation of this re-scaling by a single temperature parameter is the reduced confidence for all predictions, including the correct one. A generalization of TS is a matrix transformation which can also be learnt using a hold-out validation set. To scale the Beta-calibration (Kull et al., 2017) to a multiclass setting, Dirichlet calibration (DC) employs Dirichlet distribution (Kull et al., 2019). DC is realized as neural network layer on the log-transformed class probabilities, and the parameters are learned using a hold-out validation set. TS has shown effectiveness in calibrating in-domain predictions, but its performance suffers for out-of-domain predictions (Ovadia et al., 2019). Towards improving out-of-domain calibration, Tomani et al. (2021) perturbed the hold-out validation set prior to performing post-hoc rescaling and Yu et al. (2022) proposed multi-domain temperature scaling approach that leverages heterogeneity from multiple domains. The work of Ma & Blaschko (2021) introduced two constraints and a calibration framework comprised of a base calibrator and a ranking model. To improve calibration of networks trained on distilled data, the concurrent work of Zhu et al. (2023) proposed masked temperature scaling and masked distillation training approaches. Despite being simple and effective, several post-hoc methods assume the availability of a hold-out validation data, which is a difficult requirement to meet in several real-world applications. We approach calibration from train-time perspective, which tends to engage all model parameters during training.

**Train-time calibration approaches:** Brier score is among the earliest train-time calibration technique for verifying probabilistic forecasts (Brier et al., 1950). Decades later, Guo et al. (2017) noticed that the models trained with negative log-likelihood (NLL) tend to become overconfident, and so there is a detachment between NLL-based training and model calibration. A class of methods

proposed auxiliary loss functions which can be employed with NLL to improve model calibration. Pereyra et al. (2017) introduced a regularization term, based on entropy, to minimize the impact of overconfident predictions. Likewise, Müller et al. (2019) demonstrated that Label Smoothing (LS) is beneficial towards reducing miscalibration. Later, Mukhoti et al. (2020) showed that it is possible to implicitly calibrate the model by using Focal Loss (FL). These methods explicitly or implicitly maximise the entropy of the predictive distribution to improve model calibration, however, it could lead to degenerate solutions. To this end, Liu et al. (2022) proposed to introduce a margin between the logit distances to achieve balance between discriminative and calibration performance. Recently, Liu et al. (2023) proposes class-adaptive label smoothing method by introducing class-wise multipliers instead of single balancing weight with the penalty function. Inspired by knowledge distillation framework, Yun et al. (2020) reduced the model overfitting by penalizing the predictive distribution between similar examples. In mixup technique, Thulasidasan et al. (2019) observed that the label mixup is an important component to achieve model calibration. Lately, Wang et al. (2023) found that mixup often yields less calibratable models compared to empirical risk minimization and decoupled data transformation from random perturbation to avoid the calibration degradation in mixup. The parallel work of Noh et al. (2023) leveraged ordinal ranking relationship between the raw and mixup augmented examples which serves as another supervisory signal to improve calibration. Some methods attempt to formulate the trainable version of expected calibration error (ECE) (Naeini et al., 2015). Liang et al. (2020) introduced an auxiliary loss term called the Difference between Confidence and Accuracy (DCA) to be used with the cross-entropy loss. Similarly, Kumar et al. (2018) developed an auxiliary loss term (MMCE) using a reproducing kernel in a Hilbert space to calibrate model predictions. However, these methods only consider the maximum class confidence for calibration. To address this limitation, Hebbalaguppe et al. (2022) proposed an auxiliary loss term to calibrate both maximum and non-maximum class confidences. The concurrent work of Park et al. (2023) unifies the pros of two existing classes of train-time calibration methods. We also pursue the train-time calibration paradigm, and propose an auxiliary loss that aims to minimize, for each class, the discrepancy between the distribution of confidence scores across the mini-batch and the corresponding gradually softened ground truth occurrences distribution across the minibatch.

## 3 PROPOSED METHOD

**Preliminaries:** We consider a classification task where we are provided with a labelled dataset $\mathcal{D} = \langle (\mathbf{x}_n, y_n^*) \rangle_{n=1}^N$ comprising of $N$ examples that are sampled from a joint distribution $\mathcal{P}(\mathcal{X}, \mathcal{Y})$, where $\mathcal{X}$ is an input space, and $\mathcal{Y}$ is the label space. The tensor $\mathbf{x}_n \in \mathcal{X} \in \mathbb{R}^{H \times W \times C}$ is an input image of height $H$, width $W$, and number of channels $C$. Each input image has a corresponding ground truth class label $y_n^* \in \mathcal{Y} = \{1, 2, ..., K\}$. Given a classifier $\mathcal{F}_{cls}$, that produces a confidence vector $\mathbf{s}_n \in \mathbb{R}^K$, we regard each element of this vector $\mathbf{s}_n$ as the (predicted) confidence score of the corresponding class label. The predicted class label $\hat{y}_n$ is then given by: $\hat{y}_n = \underset{y \in \mathcal{Y}}{\arg\max}\ \mathbf{s}_n[y]$. The confidence score of the predicted class $\hat{y}_n$ is obtained as: $\hat{s}_n = \underset{y \in \mathcal{Y}}{\max}\ \mathbf{s}_n[y]$.

**Calibration:** A perfect calibration is achieved if, for a given confidence score, the (classification) accuracy is aligned with this confidence score for all possible confidence scores (Guo et al., 2017): $\mathbb{P}(\hat{y} = y^* | \hat{s} = s) = s \quad \forall s \in [0, 1]$, where $\mathbb{P}(\hat{y} = y^* | \hat{s} = s)$ is the accuracy for a given confidence score $\hat{s}$. Note that, this relation only takes into account the calibration of the predicted label corresponding to the maximum class confidence score $\hat{s}$. For completeness, the confidence score of *all* classes should also be calibrated: $\mathbb{P}(y = y^* | \mathbf{s}[y] = s) = s \quad \forall s \in [0, 1]$.

### 3.1 MEASURING MISCALIBRATION

We now discuss commonly used evaluation metrics to quantify the miscalibration of a model, namely expected calibration error (ECE), and static calibration error (SCE).

**Expected calibration error (ECE):** ECE is quantified as follows: first, for examples predicted with a specific confidence score, we calculate the absolute difference between the average predicted confidence and the average accuracy; then this difference is scaled with the fraction of examples (out of total examples in the provided set) with this specific confidence score, and finally we repeat the aforementioned steps for all possible confidence scores and take a sum (Naeini et al., 2015):

$\text{ECE} = \sum_{i=1}^{M} \frac{|B_i|}{N} \left| \frac{1}{|B_i|} \sum_{j \in B_i} \mathbb{I}(\hat{y}_j = y_j^*) - \frac{1}{|B_i|} \sum_{j:\hat{s}_j \in B_i} \hat{s}_j \right|$, where $N$ denotes the total number of examples in the provided set. As such, the confidence values have a continuous interval, so the confidence range $[0, 1]$ is partitioned into $M$ bins. $|B_i|$ is the number of examples falling in $i^{th}$ confidence bin. $\frac{1}{|B_i|} \sum_{j \in B_i} \mathbb{I}(\hat{y}_j = y_j^*)$ denotes the average accuracy of examples falling in the $i^{th}$ bin, and $\frac{1}{|B_i|} \sum_{j:\hat{s}_j \in B_i} \hat{s}_j$ denotes the average prediction confidence of examples in the $i^{th}$ confidence bin. The ECE metric is limited as it does not include the whole confidence vector, and it is not differentiable due to the partitioning of the confidence range.

**Static calibration error (SCE):** SCE generalizes ECE as it accounts for the whole confidence vector. As a result, it measures the calibration performance of non-maximum class confidences (Nixon et al., 2019): $\text{SCE} = \frac{1}{K} \sum_{i=1}^{M} \sum_{j=1}^{K} \frac{|B_{i,j}|}{N} \left| A_{i,j} - E_{i,j} \right|$, where $K$ represents the number of classes and $|B_{i,j}|$ is the number of examples from the $j^{th}$ class and the $i^{th}$ bin. $A_{i,j} = \frac{1}{|B_{i,j}|} \sum_{k \in B_{i,j}} \mathbb{I}(j = y_k)$ denotes the average accuracy and $E_{i,j} = \frac{1}{|B_{i,j}|} \sum_{k:\mathbf{s}_k[j] \in B_{i,j}} \mathbf{s}_k[j]$ represents the average confidence of the examples belonging to the $j^{th}$ class and the $i^{th}$ bin. Like ECE, SCE metric is non-differentiable, and so it cannot be used as a loss function in gradient-based optimization methods.

## 3.2 Proposed Auxiliary Loss: MACSO

We propose a novel auxiliary loss for train-time multiclass calibration. The loss formulation is inspired by the intuition that as training goes, a model's prediction becomes refined, and thus the predicted confidence scores can be gradually combined with the ground truth, to form a smoothed target distribution which has an increased entropy compared to the the one-hot encoded hard targets, potentially leading to a better calibrated model. We further argue that for a class, this gradually softened ground truth occurrences distribution is a suitable non-zero entropy signal whose better alignment with the predicted confidences distribution is positively correlated with reducing the model calibration error. In the following, we provide the detail of the proposed method, **M**ulticlass **A**lignment of **C**onfidences and **S**oftened ground truth **O**ccurrences (MACSO).

**Targets softening:** Softening one-hot encoded target vectors (i.e., hard targets) can obtain more informative labels. For example, Label Smoothing (LS) (Szegedy et al., 2016; Liu et al., 2022) results in a softened target distribution that has an increased entropy and leads to a better calibrated model. Unlike prior works whose softened targets remain unaltered throughout training, we gradually soften the hard targets via a progressive self-knowledge distillation perspective (Kim et al., 2021). Our approach is based on the intuition that a model becomes a teacher itself as training progresses, and we can progressively distill a model's own knowledge to soften hard targets during training. Specifically, targets are softened adaptively by combining the one-hot ground-truth label $\mathbf{y} \in \{0, 1\}^K$ and the predicted confidence vector $\mathbf{s}$ from the model in the last epoch:

$$\tilde{\mathbf{y}}_t = \begin{cases} \mathbf{y} & \text{if } t = 1 \\ (1 - \alpha_t)\mathbf{y} + \alpha_t \mathbf{s}_{t-1} & \text{otherwise} \end{cases} \tag{1}$$

where the subscript $t$ indexes the training epoch ($1 \leq t \leq T$), $T$ denotes the total number of training epochs, $\alpha_t = \alpha_{max} \frac{t}{T}$, and $\alpha_{max} = 0.8$ is a pre-defined hyper-parameter that controls the maximal relative ratio of predictions to be linearly combined with ground truth. $\tilde{\mathbf{y}}_t$ represents the softened targets at epoch $t$. At epoch $t$, a model is trained using $\tilde{\mathbf{y}}_t$ (instead of $\mathbf{y}$) as the label *for our auxiliary calibration loss component*. Eq. 1 shows targets are gradually softened by utilizing more information from the predicted confidence vector over the ground truth hard targets as training progresses. This target softening strategy better adapts to the progress of model training (see Fig. 8), which makes it easier to positively impact model calibration compared to prior works (e.g., Liu et al. (2022)).

**Multiclass alignment of confidences and softened targets:** We aim at the calibration of all of the $K$ classes, which is measured by the SCE metric. As SCE encourages a class-wise conformity between the ground truth occurrences and predicted confidences, we propose a differentiable auxiliary loss formulation $\mathcal{L}_{\text{MACSO}}$ that encourages the alignment between the gradually softened distribution of its ground truth occurrences and the distribution of its predicted confidences. Intuitively, this alignment is positively correlated with reducing the model calibration error. Besides simply align-

ing the two distributions, e.g., via their means or KL divergence, we propose to quantify the linear correlation between the two distributions which preserves the relations among them, thereby further improving the calibration performance. We verify empirically that correlation outperforms KL divergence (see Sec. 4). Our novel auxiliary loss is given by Eq. 2:

$$\mathcal{L}_{\text{MACSO}} = \frac{1}{K} \sum_{j=1}^{K} \left(1 - r(\mathbf{s}[j], \tilde{\mathbf{y}}[j])\right) \tag{2}$$

where $\mathbf{s}[j] \in \mathbb{R}^{N_b}$ denotes the confidence vector of the $j^{th}$ class in the mini-batch with $N_b$ training samples, $\tilde{\mathbf{y}}[j] \in \mathbb{R}^{N_b}$ is the softened target vector of the $j^{th}$ class in this mini-batch [1], and $r(\cdot, \cdot) \in [-1, 1]$ is the empirical Pearson correlation coefficient. A value of $r(\cdot, \cdot) = 1$ is the desired case which indicates a perfect alignment. The total loss $\mathcal{L}_{\text{total}}$ to train a model is described in Eq. 3:

$$\mathcal{L}_{\text{total}} = \begin{cases} \mathcal{L}_{\text{C}} & \text{if } t = 1 \\ \mathcal{L}_{\text{C}} + \beta \mathcal{L}_{\text{MACSO}} & \text{otherwise} \end{cases} \tag{3}$$

where $\mathcal{L}_{\text{C}}$ is a classification loss, such as Cross Entropy (CE/NLL) or Focal Loss (FL) (Lin et al., 2017), and $\beta$ is a pre-defined weighting hyper-parameter to control the relative importance of our novel auxiliary loss with respect to the task-specific classification loss.

*The intuitive workings of MACSO*: It is beneficial to soften hard targets for calibration because it progressively distills model's own knowledge and provides an increased entropy signal which can lead to better calibration. Since MACSO is a train-time auxiliary loss, employing softened target distribution for the calibration loss term only allows to preserve the original classification task while optimizing the model for calibration. Additionally, in multiclass calibration, we care about preserving the class relations. Pearson correlation based loss function allows the model to be guided appropriately to distill those truly informative multiclass relations. Pearson correlation underscores linear relationships between softened ground truth occurrence of class $i$ and predicted confidence score of class $i$. It captures the intrinsic inter-class relations, making it superior to KL divergence for multi-class calibration by penalizing less when the two distributions differ only in class absolute scores but multi-class relations remain preserved. We provide extended discussions in the Appendix. We observe from our analysis (Fig. 2) that, compared to aligning the confidence distribution with the hard target occurrences distribution (Hebbalaguppe et al., 2022), our proposal of using gradually softened target occurrences coupled with measuring correlation-based distance has a stronger correlation between the calibration loss and the calibration error.

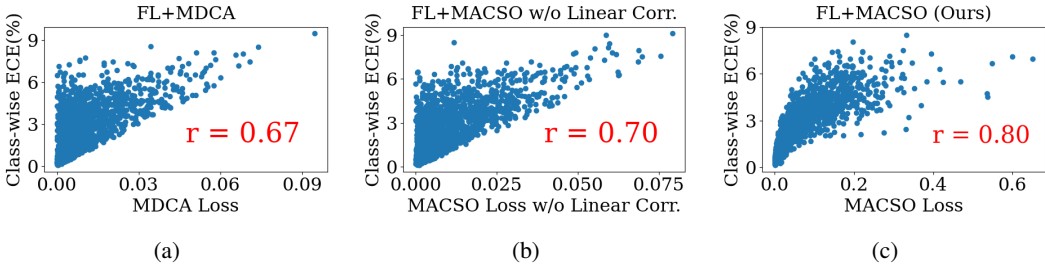

| (a) | (b) | (c) |

Figure 2: We plot relationship (via measuring correlation coefficient $r$) between class-wise ECE and class-wise auxiliary calibration losses: (a) the MDCA loss (Hebbalaguppe et al., 2022), (b) MACSO without linear correlation—where gradually softened target occurrences are used and the mean of distributions are aligned, and (c) MACSO. Instead of hard targets occurrences (a), upon aligning the distributions and using gradually softened target occurrences for the calibration loss term (b), the correlation to class-wise ECE improves. After further leveraging linear correlation for alignment (c), this correlation between calibration loss and calibration metric improves further and significantly. Refer to Appendix for implementation details of this figure.

## 3.3 THEORETICAL PROPERTIES OF MACSO

In this section, we show that MACSO has desirable theoretical properties. Proofs for our results can be found in Appendix A. For simplicity of notation, we will consider a binary (rather than multi-

---

[1]Hereon and after, we omit the subscript $t$ representing the current epoch for readability.

class) classification task, but all of the results generalize to the multi-class setting. First, we show that a population-level version of MACSO is (almost) a proper scoring rule. Define $\mathcal{L}_{\mathrm{MACSO-Pop}}$ via the population (rather than empirical) correlation:

$$\mathcal{L}_{\mathrm{MACSO-Pop}} = 1 - \frac{\mathbb{E}[(\mathbf{s} - \mathbb{E}\mathbf{s})(\tilde{\mathbf{y}} - \mathbb{E}\tilde{\mathbf{y}})]}{\sqrt{\mathbb{E}[(\mathbf{s} - \mathbb{E}\mathbf{s})^2]\mathbb{E}[(\tilde{\mathbf{y}} - \mathbb{E}\tilde{\mathbf{y}})^2]}} = 1 - \mathrm{corr}(\mathbf{s}, \tilde{\mathbf{y}}). \tag{4}$$

Here we are overloading notation and use $\mathbf{s}$ to refer to the random variable $\mathbf{s}(\mathbf{x})$, i.e., the output of our model when it is applied to a sample $\mathbf{x}$ drawn from the data distribution. Similarly, $\tilde{\mathbf{y}}$ is the random variable $\alpha\mathbf{s}(\mathbf{x}) + (1-\alpha)\mathbf{y}$, where $(\mathbf{x}, \mathbf{y})$ are drawn from the data distribution.

By a proper scoring rule, we mean a loss function which is minimized by predicting $\mathbb{E}[\mathbf{y}|\mathbf{x}]$. If we consider optimization of the loss *jointly* over $\mathbf{s}$ and $\tilde{\mathbf{y}}$, then $\mathcal{L}_{\mathrm{MACSO-Pop}}$ cannot be a proper scoring rule. This is because when $\alpha > 0$, $\mathrm{corr}(\mathbf{s}, \alpha\mathbf{s} + (1-\alpha)\mathbf{y})$ can be increased by having the variability in the $\mathbf{s}$ term dominate the variability in $\mathbf{y}$. (For instance, if there were no restrictions placed on $\mathbf{s}$, we could have $\mathbf{s}$ be highly variable in $\mathbf{x}$ with magnitude $\gg$ the magnitude of $\mathbf{y}$; this clearly improves the correlation between $\mathbf{s}(\mathbf{x})$ and $\tilde{\mathbf{y}}$.) Instead, we consider the following procedure. Given some initialization for $\mathbf{s}$, define the corresponding random variable $\tilde{\mathbf{y}}$. Treating this $\tilde{\mathbf{y}}$ as fixed, update $\mathbf{s}$ as the argmin of $\mathcal{L}_{\mathrm{MACSO-Pop}}$, i.e. update $\mathbf{s}$ so that it minimizes $1 - \mathrm{corr}(\mathbf{s}, \tilde{\mathbf{y}})$ treating $\tilde{\mathbf{y}}$ as fixed. Using this new $\mathbf{s}$, update the definition of the random variable $\tilde{\mathbf{y}}$ and repeat the procedure. We will say that $\mathcal{L}_{\mathrm{MACSO-Pop}}$ is a pseudo-proper scoring rule if $\mathbf{s}(\mathbf{x}) = \mathbb{E}[\mathbf{y}|\mathbf{x}]$ is a fixed point of this procedure. Theorem 1 shows that this is indeed the case.

**Theorem 1.** $\mathcal{L}_{\mathrm{MACSO-Pop}}$ *is a pseudo proper scoring rule. That is,* $\mathbf{s}(\mathbf{x}) = \mathbb{E}[\mathbf{y}|\mathbf{x}]$ *is a fixed point of the procedure whereby we treat* $\tilde{\mathbf{y}}$ *as fixed, optimize* $\mathcal{L}_{\mathrm{MACSO-Pop}}$ *with respect to the fixed* $\tilde{\mathbf{y}}$, *recompute* $\tilde{\mathbf{y}}$ *using the updated* $\mathbf{s}$, *and repeat.*

Next, we consider the learning dynamics induced by the loss in the finite data regime. We will show that the gradient of the MACSO loss induces an implicit regularization effect during learning which discourages overconfidence in the model predictions. We define $\bar{\tilde{y}} = \frac{1}{N_b}\sum_{1 \leq i \leq N_b} \tilde{\mathbf{y}}_i$ and $\bar{s} = \frac{1}{N_b}\sum_{1 \leq i \leq N_b} \mathbf{s}_i$, and we let $\bar{\mathbf{s}}$ and $\bar{\tilde{\mathbf{y}}}$ be the vectors of length $N_b$ whose entries are all $\bar{s}$ or $\bar{\tilde{y}}$, respectively. Let $\Delta\mathbf{s} = \mathbf{s} - \bar{\mathbf{s}}$ and $\Delta\tilde{\mathbf{y}} = \tilde{\mathbf{y}} - \bar{\tilde{\mathbf{y}}}$. It can be shown that

$$-\nabla\mathcal{L}_{\mathrm{MACSO}} \propto \sum_{i=1}^{N_b} \left( \underbrace{\|\Delta\mathbf{s}\|\|\Delta\tilde{\mathbf{y}}\|(\tilde{\mathbf{y}}_i - \bar{\tilde{y}})}_{(\mathrm{I})} - \underbrace{\frac{(\Delta\mathbf{s} \cdot \Delta\tilde{\mathbf{y}})\|\Delta\tilde{\mathbf{y}}\|}{\|\Delta\mathbf{s}\|}(\mathbf{s}_i - \bar{s})}_{(\mathrm{II})} \right) \nabla\mathbf{s}_i. \tag{5}$$

That is, $-\nabla\mathcal{L}_{\mathrm{MACSO}}$ is a positive scalar times the expression in Eq. 5. In term (I), $\|\Delta\mathbf{s}\|$ and $\|\Delta\tilde{\mathbf{y}}\|$ are always positive. Thus, this term will encourage *increasing* $\mathbf{s}_i$ if $\tilde{\mathbf{y}}_i$ is large or *decreasing* $\mathbf{s}_i$ if $\tilde{\mathbf{y}}_i$ is small. This is similar to the effect of the cross-entropy loss, i.e., encouraging the model to make predictions close to the specified labels. In term (II), we may assume that in the later stages of training, the confidence scores and (pseudo-) labels are fairly well aligned. In this case, we should have $\Delta\mathbf{s} \cdot \Delta\tilde{\mathbf{y}} > 0$. Since $\|\Delta\tilde{\mathbf{y}}\|$ and $\|\Delta\mathbf{s}\|$ are always positive, term (II) will tend to *decrease* $\mathbf{s}_i$ if $\mathbf{s}_i > \bar{s}$ and *increase* $\mathbf{s}_i$ if $\mathbf{s}_i < \bar{s}$. Thus, term (II) can be seen as providing a regularizing effect, encouraging *more uniform confidence predictions across samples*. In particular, this should discourage the model from making extremely confident predictions and improve calibration.

## 4 EXPERIMENTS

**Datasets:** We evaluate the in-domain calibration performance of our method rigourously with four benchmark image classification datasets: CIFAR10 (Krizhevsky et al., 2009), CIFAR100 (Krizhevsky et al., 2009), Tiny-ImageNet (Deng et al., 2009) and Mendeley V2 (Kermany et al., 2018). In addition, to validate out-of-domain calibration performance, we use three benchmark datasets: CIFAR10-C, CIFAR100-C and Tiny-ImageNet-C. Moreover, we validate our method on SVHN (Netzer et al., 2011) to report calibration performance under class imbalance.

**Evaluation metrics and baselines:** We report the calibration performance with metrics SCE (Nixon et al., 2019), ECE (Naeini et al., 2015) and classification performance with the top-1 accuracy. We

report the mean and standard deviation of all metrics over 10 trials. We set the number of bins ($M$) = 15. To visualize the calibration performance, we plot reliability diagrams and confidence histograms. We compare our method against models trained with NLL, FL (Lin et al., 2017), LS (Müller et al., 2019), BS (Brier et al., 1950) and FLSD (Mukhoti et al., 2020). In addition, we draw comparisons with existing auxiliary loss functions for calibration such as MMCE (Kumar et al., 2018), MbLS (Liu et al., 2022) and MDCA (Hebbalaguppe et al., 2022). All hyper-parameters for the baselines are set according to the values reported in the original works. We use ResNet (Wu et al., 2019) as the backbone in all our experiments. Please refer to Appendix for more implementation details.

**Experiments with task-specific loss functions:** MACSO is designed to be used with different task-specific losses. We use NLL or FL as the task-specific loss and report results with and without MACSO. Table 1 shows that our MACSO delivers consistent gains over the two task-specific losses across all five datasets in both of the SCE and ECE metrics. FL tends to be a strong task-specific loss in calibration performance, except in SVHN. Therefore, in all subsequent experiments, we report performance with FL+MACSO on all datasets, except SVHN (for which we use NLL loss).

| Dataset | NLL | | NLL+MACSO (ours) | | FL (Lin et al., 2017) | | FL+MACSO (ours) | |
|---|---|---|---|---|---|---|---|---|
| | SCE ↓ | ECE ↓ | SCE ↓ | ECE ↓ | SCE ↓ | ECE ↓ | SCE ↓ | ECE ↓ |
| CIFAR10 | $7.04 \pm 0.30$ | $3.19 \pm 0.18$ | $6.52 \pm 0.46$ | $2.86 \pm 0.26$ | $\underline{3.98 \pm 0.30}$ | $\underline{1.06 \pm 0.26}$ | $\mathbf{3.88 \pm 0.21}$ | $\mathbf{1.06 \pm 0.22}$ |
| CIFAR100 | $2.61 \pm 0.04$ | $9.17 \pm 0.31$ | $\mathbf{1.80 \pm 0.04}$ | $\mathbf{1.36 \pm 0.25}$ | $1.96 \pm 0.07$ | $1.73 \pm 0.78$ | $\underline{1.82 \pm 0.05}$ | $\underline{1.43 \pm 0.20}$ |
| Tiny-ImageNet | $2.09 \pm 0.08$ | $14.24 \pm 1.04$ | $1.87 \pm 0.03$ | $11.57 \pm 0.31$ | $\underline{1.50 \pm 0.02}$ | $\underline{3.32 \pm 0.45}$ | $\mathbf{1.44 \pm 0.02}$ | $\mathbf{1.65 \pm 0.27}$ |
| SVHN | $2.39 \pm 0.56$ | $\underline{0.61 \pm 0.42}$ | $\mathbf{2.17 \pm 0.22}$ | $\mathbf{0.53 \pm 0.15}$ | $6.05 \pm 2.44$ | $2.69 \pm 1.43$ | $5.77 \pm 0.54$ | $2.62 \pm 0.30$ |
| Mendeley | $236 \pm 17.9$ | $18.69 \pm 2.09$ | $225 \pm 21.7$ | $16.82 \pm 2.55$ | $\underline{222 \pm 14.3}$ | $\underline{15.18 \pm 2.03}$ | $\mathbf{205 \pm 23.0}$ | $\mathbf{13.71 \pm 2.56}$ |
| CIFAR10 (soft)* | $\underline{4.49 \pm 0.39}$ | $\underline{1.37 \pm 0.24}$ | $\mathbf{4.45 \pm 0.51}$ | $\mathbf{1.28 \pm 0.24}$ | $63.9 \pm 6.30$ | $32.53 \pm 3.35$ | $60.5 \pm 8.35$ | $29.98 \pm 4.04$ |

Table 1: Calibration performance in SCE ($10^{-3}$) and ECE (%) of our MACSO loss when added to two task-specific losses: NLL and FL. Best results of each dataset are in bold, and the second best are underlined. MACSO consistently delivers gains over the task-specific loss and achieves the best results across all cases. Refer to Appendix for classification accuracies.

We additionally report the results when the gradually softened targets (i.e., Eq 1) are utilized as a mere replacement for hard targets within a conventional task-specific classification loss, in the last row of Table 1. In this case, NLL/FL is instead termed as $NLL_{soft}$/$FL_{soft}$. Although $NLL_{soft}$ has improved performance over NLL, same is not observed with FL. This confirms the calibration efficacy of gradually softened targets employed within the calibration loss component. Notably, MACSO (i.e., $NLL_{soft}$/$FL_{soft}$+MACSO) provides gains over both $NLL_{soft}$ and $FL_{soft}$ as well.

**Comparison with the state-of-the-art (SOTA) methods:** Table 2 show the calibration performance of our method against previous SOTA train-time calibration methods. See Appendix for detailed justifications on the choices made for selecting task-specific losses for MDCA, MbLS and our MACSO. Our method consistently shows lower calibration errors than the competitors across all five datasets.

| Dataset | BS (Brier et al., 1950) | | MMCE (Kumar et al., 2018) | | FLSD (Mukhoti et al., 2020) | | LS (Müller et al., 2019) | |
|---|---|---|---|---|---|---|---|---|
| | SCE ↓ | ECE ↓ | SCE ↓ | ECE ↓ | SCE ↓ | ECE ↓ | SCE ↓ | ECE ↓ |
| CIFAR10 | $5.79 \pm 0.41$ | $2.33 \pm 0.21$ | $7.94 \pm 0.82$ | $3.13 \pm 0.39$ | $9.66 \pm 1.03$ | $4.42 \pm 0.55$ | $6.33 \pm 0.30$ | $1.96 \pm 0.22$ |
| CIFAR100 | $2.21 \pm 0.10$ | $5.47 \pm 0.67$ | $2.30 \pm 0.12$ | $6.32 \pm 1.09$ | $2.00 \pm 0.03$ | $1.97 \pm 0.27$ | $1.97 \pm 0.06$ | $2.86 \pm 0.56$ |
| Tiny-ImageNet | - | - | - | - | $1.49 \pm 0.02$ | $3.48 \pm 0.59$ | $1.50 \pm 0.06$ | $2.56 \pm 0.62$ |
| SVHN | $2.94 \pm 0.34$ | $0.88 \pm 0.22$ | $12.9 \pm 0.85$ | $6.39 \pm 0.43$ | $24.2 \pm 3.46$ | $12.15 \pm 1.81$ | $10.8 \pm 0.76$ | $4.79 \pm 0.34$ |
| Mendeley | $229 \pm 29.4$ | $17.76 \pm 3.62$ | $229 \pm 16.4$ | $15.49 \pm 2.53$ | $229 \pm 17.5$ | $16.35 \pm 2.44$ | $217 \pm 13.3$ | $15.97 \pm 1.75$ |

| Dataset | NLL/FL+MDCA (Hebbalaguppe et al., 2022) | | | NLL+MbLS (Liu et al., 2022) | | | NLL/FL+MACSO (ours) | | |
|---|---|---|---|---|---|---|---|---|---|
| | SCE ↓ | ECE ↓ | Acc. ↑ | SCE ↓ | ECE ↓ | Acc. ↑ | SCE ↓ | ECE ↓ | Acc. ↑ |
| CIFAR10 | $\underline{4.34 \pm 0.58}$ | $1.38 \pm 0.38$ | $92.97 \pm 0.20$ | $5.32 \pm 0.08$ | $\underline{1.93 \pm 0.19}$ | $93.39 \pm 0.15$ | $\mathbf{3.88 \pm 0.21}$ | $\mathbf{1.06 \pm 0.22}$ | $93.07 \pm 0.31$ |
| CIFAR100 | $1.98 \pm 0.04$ | $2.04 \pm 0.42$ | $72.02 \pm 0.25$ | $\underline{1.95 \pm 0.06}$ | $1.87 \pm 0.58$ | $72.94 \pm 0.55$ | $\mathbf{1.82 \pm 0.05}$ | $\mathbf{1.43 \pm 0.20}$ | $73.37 \pm 0.65$ |
| Tiny-ImageNet | $1.49 \pm 0.02$ | $3.62 \pm 0.44$ | $61.49 \pm 0.51$ | $\underline{1.47 \pm 0.03}$ | $\underline{2.49 \pm 0.30}$ | $61.22 \pm 1.06$ | $\mathbf{1.44 \pm 0.02}$ | $\mathbf{1.65 \pm 0.27}$ | $61.05 \pm 0.27$ |
| SVHN | $\underline{2.25 \pm 0.68}$ | $\underline{0.59 \pm 0.43}$ | $96.53 \pm 0.12$ | $2.48 \pm 0.38$ | $0.77 \pm 0.16$ | $96.75 \pm 0.16$ | $\mathbf{2.17 \pm 0.22}$ | $\mathbf{0.53 \pm 0.15}$ | $96.71 \pm 0.16$ |
| Mendeley | $227 \pm 19.9$ | $18.37 \pm 2.24$ | $76.28 \pm 2.22$ | $244 \pm 14.4$ | $19.66 \pm 1.98$ | $74.23 \pm 1.58$ | $\mathbf{205 \pm 23.0}$ | $\mathbf{13.71 \pm 2.56}$ | $75.82 \pm 1.51$ |

Table 2: Performance comparison with SOTA train-time calibration methods. Our method consistently demonstrates superior calibration performance compared to competitors

**Class-wise calibration performance:** The complete definition of calibration requires that the whole confidence vector should be calibrated and not just the predicted class confidence. Fig. 3 reports class-wise ECE scores of our MACSO method and the competing auxiliary-loss-based calibration approaches (i.e., MDCA and MbLS). We chose to report results based on the task-specific loss (NLL/FL) that yielded superior performance. In SVHN, NLL+MACSO (ours) achieves the lowest ECE in seven (out of ten) classes while demonstrating the second best score in the rest. In CIFAR10, FL+MACSO (ours) provides the best ECE scores in nine classes out of ten.

**Calibration performance under class imbalance:** Real-world datasets are often dominated by few classes over the rest. For benchmarking calibration performance under class imbalance, we utilize SVHN dataset, which has a class imbalance factor of 2.7 (Hebbalaguppe et al., 2022). Tables 1, 2 and Fig. 3 reveal that, in comparison to other train-time auxiliary losses, ours delivers the best ECE and SCE scores over both of the whole dataset and seven classes (out of ten) in the SVHN dataset.

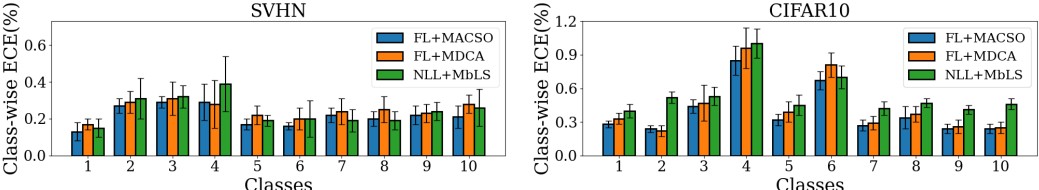

Figure 3: Class-wise calibration performance in ECE (↓) on SVHN and CIFAR10 benchmarks with ResNet56.

**Out-of-domain performance:** We report out-of-domain calibration performance on CIFAR10-C, CIFAR100-C and Tiny-ImageNet-C (Table 3). In ECE metric, our method outperforms all competing approaches in all three datasets. In SCE metric, it displays the best performance in CIFAR100-C and the second best performance in Tiny-ImageNet-C and CIFAR10-C. For more results on out-of-domain performance, refer to Fig. 12 in the Appendix.

| Dataset | FL+MDCA (Hebbalaguppe et al., 2022) | | | NLL+MbLS (Liu et al., 2022) | | | FL+MACSO | | |
|---|---|---|---|---|---|---|---|---|---|
| | SCE ($10^{-3}$)↓ | ECE (%)↓ | Acc. (%)↑ | SCE ($10^{-3}$)↓ | ECE (%)↓ | Acc. (%)↑ | SCE ($10^{-3}$)↓ | ECE (%)↓ | Acc. (%)↑ |
| CIFAR10-C | $29.41 \pm 2.62$ | $12.12 \pm 1.44$ | $72.24 \pm 0.93$ | $\mathbf{28.92 \pm 2.23}$ | $13.05 \pm 1.17$ | $73.55 \pm 0.75$ | $28.97 \pm 1.62$ | $\mathbf{11.72 \pm 0.81}$ | $72.00 \pm 0.81$ |
| CIFAR100-C | $4.81 \pm 0.21$ | $16.52 \pm 1.08$ | $44.57 \pm 0.50$ | $\underline{4.28 \pm 0.18}$ | $14.81 \pm 1.07$ | $45.79 \pm 0.52$ | $\mathbf{4.16 \pm 0.09}$ | $\mathbf{12.64 \pm 0.63}$ | $46.33 \pm 0.25$ |
| Tiny-ImageNet-C | $3.75 \pm 0.08$ | $21.41 \pm 0.50$ | $21.85 \pm 0.56$ | $\mathbf{3.27 \pm 0.16}$ | $19.00 \pm 1.19$ | $22.12 \pm 1.01$ | $\underline{3.49 \pm 0.05}$ | $\mathbf{16.19 \pm 0.59}$ | $21.69 \pm 1.03$ |

Table 3: Out-of-domain calibration performance across CIFAR10-C, CIFAR100-C and Tiny-ImageNet-C.

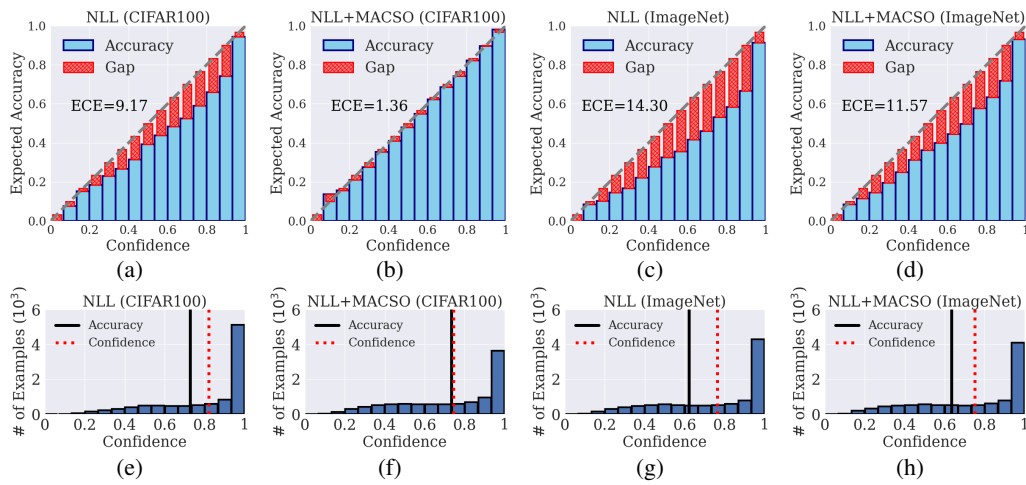

Figure 4: Reliability diagrams (a,b,c,d) and confidence histograms (e,f,g,h) of (ResNet) models trained with NLL and NLL+MACSO. Diagrams (a,b,e,f) and (c,d,g,h) reveal that MACSO can reduce over-confidence.

**Reliability diagrams and confidence histograms:** We plot reliability diagrams to visualize the calibration performance (Fig. 4 (top)). Compared to NLL, our method is capable of reducing the gap between accuracy and confidence at all levels in CIFAR100 and Tiny-ImageNet datasets. Furthermore, we plot confidence histograms to visualize the deviation between the overall confidence (dotted line) and accuracy (solid line) of the predictions (Fig. 4 (bottom)). Contrary to NLL, our method is better at reducing the gap between the overall confidence and accuracy, thereby effectively tackling the overconfident behavior. Fig. 5 further depicts the histogram of confidence values for the incorrect predictions from CIFAR100 and SVHN datasets. We note that, compared to NLL, for NLL+MACSO the confidence values of the incorrect predictions are smaller. These experimental findings (demonstrated in Fig. 4 (bottom) and Fig. 5) are in accordance with our theoretical results, i.e., the gradient of the MACSO loss induces an regularization effect during learning which discourages overconfidence in the model prediction.

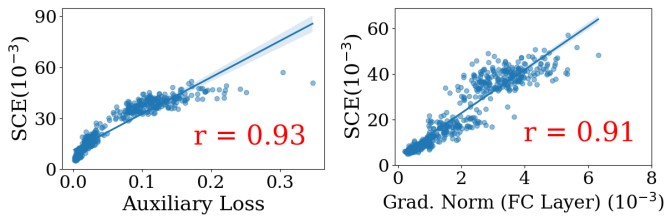

| NLL (CIFAR100) | NLL+MACSO (CIFAR100) | NLL (SVHN) | NLL+MACSO (SVHN) |
| :---: | :---: | :---: | :---: |
| (a) | (b) | (c) | (d) |

Figure 5: Histogram of confidence values for incorrect predictions from CIFAR100 and SVHN. Compared to NLL, for NLL+MACSO the confidence values of the incorrect predictions are smaller. Our proposed MACSO auxiliary loss excels in mitigating overconfidence in mis-predictions.

**Impact of different components in MACSO:** We show the performance contribution of two different components in our auxiliary loss: gradually softened target occurrences and linear correlation, in Table 4. Upon including gradually softened target occurrences (i.e., FL+MACSO (Mean)) in place of hard target occurrences (i.e., FL+MDCA), we see a significant improvement in both calibration metrics. Moreover, after employing linear correlation with softened target occurrences (i.e., FL+MACSO (Linear Corr.)), there is a notable and consistent gain over FL+MACSO (Mean). In short, both components are vital towards realizing the full potential of our method MACSO.

**Linear correlation vs. KL divergence:** Table 4 also compares the calibration performance when using KL divergence (FL+MACSO (KL)) to measure the discrepancy instead of linear correlation. We see that linear correlation provides better gains compared to KL divergence in the majority of the datasets (CIFAR100 and Tiny-ImageNet). Compared to CIFAR10, both CIFAR100 and Tiny-ImageNet are relatively challenging and large-scale.

| Dataset | FL+MDCA (Hebbalaguppe et al., 2022) | | | FL+MACSO (Mean) | | | FL+MACSO (Linear Corr.) | | | FL+MACSO (KL) | | |
| :--- | :---: | :---: | :---: | :---: | :---: | :---: | :---: | :---: | :---: | :---: | :---: | :---: |
| | SCE $(10^{-3})\downarrow$ | ECE (%) $\downarrow$ | Acc. (%) $\uparrow$ | SCE $(10^{-3})\downarrow$ | ECE (%) $\downarrow$ | Acc. (%) $\uparrow$ | SCE $(10^{-3})\downarrow$ | ECE (%) $\downarrow$ | Acc. (%) $\uparrow$ | SCE $(10^{-3})\downarrow$ | ECE (%) $\downarrow$ | Acc. (%) $\uparrow$ |
| CIFAR10 | $4.3 \pm 0.6$ | $1.4 \pm 0.4$ | $93.0 \pm 0.2$ | $4.0 \pm 0.3$ | $1.2 \pm 0.2$ | $93.0 \pm 0.3$ | $\underline{3.9 \pm 0.2}$ | $\underline{1.1 \pm 0.2}$ | $93.0 \pm 0.3$ | $\mathbf{3.8 \pm 0.4}$ | $\mathbf{0.9 \pm 0.2}$ | $93.0 \pm 0.3$ |
| CIFAR100 | $2.0 \pm 0.0$ | $2.0 \pm 0.4$ | $72.0 \pm 0.3$ | $2.0 \pm 0.0$ | $1.9 \pm 0.4$ | $71.9 \pm 0.5$ | $\mathbf{1.8 \pm 0.1}$ | $\mathbf{1.4 \pm 0.2}$ | $73.4 \pm 0.7$ | $\underline{1.9 \pm 0.0}$ | $\underline{1.8 \pm 0.4}$ | $72.3 \pm 0.4$ |
| Tiny-ImageNet | $1.5 \pm 0.0$ | $3.6 \pm 0.4$ | $61.5 \pm 0.5$ | $1.5 \pm 0.0$ | $3.2 \pm 0.3$ | $61.1 \pm 0.3$ | $\mathbf{1.4 \pm 0.0}$ | $\underline{1.7 \pm 0.3}$ | $61.1 \pm 0.3$ | $\underline{1.5 \pm 0.0}$ | $2.8 \pm 0.4$ | $61.3 \pm 0.7$ |

Table 4: Impact of different components and linear correlation vs KL divergence in our method (MACSO). Although merely aligning softened targets and ground truth occurrences distributions using methods such as mean alignment or KL divergence offers improved model calibration compared to prior methods, our final method takes it a step further. We delve into the measurement of their linear correlation, preserving the relative importance and order, which, as our results indicate, offers superior calibration improvement.

**Relationship between MACSO and SCE:** Figure 6 illustrates a significant correlation between SCE and both our auxiliary loss (left) and gradients propagated with our auxiliary loss (right).

Figure 6: There exists strong correlation between the multiclass calibration metric and our multiclass auxiliary loss (left) and the magnitude of gradients (at fully connected (FC) layer) backpropagating with our auxiliary loss (right). For results in this figure, ResNet56 is trained on CIFAR10 with FL+MACSO.

## 5 CONCLUSION

We present a new train-time calibration method, MACSO, featuring an auxiliary loss formulation that achieves multiclass alignment of confidence distribution and the corresponding distribution of gradually softened target occurrences. Besides aligning the two distributions, we propose to measure the linear correlation between them. Empirical results shows that our loss is strongly correlated to the calibration metrics. Extensive experiments on challenging benchmarks, exhibiting in-domain, out-of-domain, and class-imbalance scenarios alongside medical imaging classification task, corroborate the efficacy of our method against the established train-time calibration methods. Moreover, MACSO has desirable theoretical properties, which help explain MACSO's performance.

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
