# Appendix

# A  THEOREMS AND PROOFS

**Lemma 2.** *For any random variables $X, Y$, it is always the case that $f(X) = \mathbb{E}[Y|X]$ maximizes* $\mathrm{corr}(f(X), Y)$ *over all choices of $f$.*

*Proof.* It is a standard result that for any two random variables $Z_1, Z_2$, $\mathrm{corr}(Z_1, Z_2)$ is maximized if $Z_1 = cZ_2$ for some positive scalar $c$. In particular, this means that $f(X) = \mathbb{E}[Y|X]$ maximizes $\mathrm{corr}(f(X), \mathbb{E}[Y|X])$ over all choices of $f$. Next, observe that

$$
\begin{aligned}
\mathrm{corr}(f(X), Y) &= \frac{\mathbb{E}[(f(X) - \mathbb{E}f)(Y - \mathbb{E}Y)]}{\sqrt{\mathbb{E}[(f(X) - \mathbb{E}f)^2]\mathbb{E}[(Y - \mathbb{E}Y)^2]}} \\
&= \frac{\mathbb{E}[(f(X) - Ef)(\mathbb{E}[Y|X] - \mathbb{E}Y)]}{\sqrt{\mathbb{E}[(f(X) - \mathbb{E}f)^2]\mathbb{E}[(\mathbb{E}[Y|X] - \mathbb{E}Y)^2]}} \cdot \frac{\sqrt{\mathbb{E}[(\mathbb{E}[Y|X] - \mathbb{E}Y)^2]}}{\sqrt{\mathbb{E}[(Y - \mathbb{E}Y)^2]}} \quad (6) \\
&= \mathrm{corr}(f(X), \mathbb{E}[Y|X]) \cdot \underbrace{\frac{\sqrt{\mathbb{E}[(\mathbb{E}[Y|X] - \mathbb{E}Y)^2]}}{\sqrt{\mathbb{E}[(Y - \mathbb{E}Y)^2]}}}_{\text{positive constant w.r.t. } f} .
\end{aligned}
$$

Eq. 6 follows from an application of the tower rule. In particular, we see that $\mathrm{corr}(f(X), Y)$ and $\mathrm{corr}(f(X), \mathbb{E}[Y|X])$ are related by a positive constant factor. Since $f(X) = \mathbb{E}[Y|X]$ maximizes the latter, it also maximizes the former, as desired. $\square$

**Theorem 1.** $\mathcal{L}_{\mathrm{MACSO-Pop}}$ *is a pseudo proper scoring rule. That is, $\mathbf{s}(\mathbf{x}) = \mathbb{E}[\mathbf{y}|\mathbf{x}]$ is a fixed point of the procedure whereby we treat $\tilde{\mathbf{y}}$ as fixed, optimize $\mathcal{L}_{\mathrm{MACSO-Pop}}$ with respect to the fixed $\tilde{\mathbf{y}}$, recompute $\tilde{\mathbf{y}}$ using the updated $\mathbf{s}$, and repeat.*

*Proof.* When $\mathbf{s}(\mathbf{x}) = \mathbb{E}[\mathbf{y}|\mathbf{x}]$, we have $\tilde{\mathbf{y}} = \alpha \mathbb{E}[\mathbf{y}|\mathbf{x}] + (1 - \alpha)\mathbf{y}$. By the Lemma 2, $\mathrm{corr}(\mathbf{s}(\mathbf{x}), \tilde{\mathbf{y}})$ is maximized if $\mathbf{s}(\mathbf{x}) = \mathbb{E}[\tilde{\mathbf{y}}|\mathbf{x}]$. This equality holds when $\mathbf{s}(\mathbf{x}) = \mathbb{E}[\mathbf{y}|\mathbf{x}]$, because

$$
\mathbb{E}[\tilde{\mathbf{y}}|\mathbf{x}] = \mathbb{E}[\alpha \mathbf{s}(\mathbf{x}) + (1 - \alpha)\mathbf{y}|\mathbf{x}] = \alpha \mathbf{s}(\mathbf{x}) + (1 - \alpha)\mathbb{E}[\mathbf{y}|\mathbf{x}] = \alpha \mathbf{s}(\mathbf{x}) + (1 - \alpha)\mathbf{s}(\mathbf{x}) = \mathbf{s}(\mathbf{x}).
$$

This completes the proof. $\square$

# B  MOTIVATION FOR MACSO DESIGN CHOICES

## B.1  MOTIVATION ON UTILIZING TARGET SOFTENING FOR MACSO

Intuitively, a model becomes a teacher itself as training progresses. Therefore, it can be beneficial to progressively distill a model's own knowledge during training. We propose to progressively distill a model's own knowledge to soften hard targets for particularly the calibration auxiliary loss, because it is also empirically confirmed that a softened target distribution has an increased entropy, which can lead to a better calibrated model. Since our proposed method is a train-time auxiliary loss for calibration, we suggest employing the softened target distribution exclusively for the calibration loss term, ensuring that the classification task remains unaffected so that classification performance can be maximized while the model is also optimized for calibration. This proposed target softening strategy for calibration better adapts to the progress of model training, which makes it easier to positively impact models compared to prior works.

## B.2  MOTIVATION ON FAVORING PEARSON CORRELATION OVER KL DIVERGENCE

We aim for the increase (or decrease) in the softened occurrence of a random ground truth class $i$ to be associated with the increase (or decrease) in the predicted confidence of that same class $i$. Pearson correlation is particularly useful in this context as it emphasizes the linear trend in the joint distribution of two variables. It is better suited than KL divergence for capturing such linear relationships between the ground truth class occurrences and predicted class confidences. This holds especially true in the context of calibrating multi-class confidence scores, where we actually hope to preserve the multi-class relations. Given that the Pearson correlation coefficient quantifies the

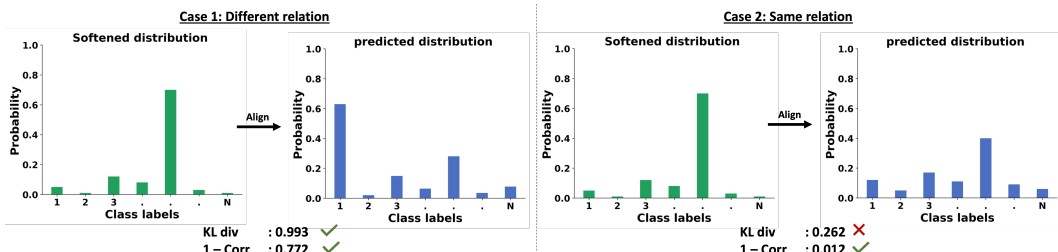

Figure 7: KL vs correlation loss behaviour.

strength and direction of a linear relationship, Pearson correlation can better ensure the alignment between the (softened) ground truth class occurrences and predicted class confidences in terms of the relations of multiple classes. When there's a discrepancy in both class scores and class relations, both KL divergence and Pearson correlation based losses would penalize the model. However, we argue that the KL divergence loss is sub-optimal because it can unnecessarily penalize the model when the relations of classes are the same. Therefore, KL divergence would disturb the training for multi-class calibration.

To this end, we illustrate two different cases in Fig. 7: different relation (left) and same relation (right) between the (softened) ground truth, which leverages the model's prediction from the previous epoch, and the current model prediction. For the KL divergence loss, the model faces undue penalties when class relations remain consistent, but scores differ marginally. On the other hand, a relation preserving loss, such as the Pearson correlation loss $(1 - \mathrm{corr})$, only (strongly) penalizes the model when both of the class score and relation are different. We also provide a theoretical intuition comparing the KL divergence loss and the Pearson correlation loss. Let us assume, without loss of generality, that we approximate the confidence distribution as the normal distribution. Specifically, we take the softened ground truth distribution as $\mathcal{N}(c_L, s_L^2)$ and the current prediction distribution as $\mathcal{N}(c_I, s_I^2)$, where $c_L$ and, $c_I$ are the mean of the distributions and $s_L$, and $s_I$ are the standard deviation. The KL divergence as loss can be computed as:

$$\mathrm{KL} = \log\left(\frac{s_L}{s_I}\right) + \frac{s_I^2 + (c_I - c_L)^2}{2s_L^2} - \frac{1}{2}. \tag{7}$$

From this equation, we can see that even if the means are equal, and only the standard deviation differs, the loss will be significant, especially if the $s_I \gg s_L$. However, the correlation will always be 1 if the means $c_I$ and $c_L$ are the same and the correlation loss, i.e., $1 - \mathrm{corr}$ is 0. Therefore, when the relation/order is same, the Pearson correlation loss is approximately 0 as opposed to KL divergence loss.

## C   ADDITIONAL QUANTITATIVE COMPARISONS

### C.1   COMPARISON ON CLASSIFICATION PERFORMANCE

Table 5 reports the classification accuracies of SOTA train-time calibration methods. We note that, our MACSO demonstrates competitive accuracy against the competing methods.

| Dataset | NLL | LS | FL | BS | MMCE | FLSD | NLL+MACSO | FL+MACSO |
|---------|-----|-----|-----|-----|------|------|-----------|----------|
| CIFAR10 | $93.19 \pm 0.23$ | $\mathbf{93.49 \pm 0.21}$ | $92.94 \pm 0.19$ | $92.31 \pm 0.21$ | $91.69 \pm 0.22$ | $92.30 \pm 0.23$ | $\underline{93.29 \pm 0.22}$ | $93.07 \pm 0.31$ |
| CIFAR100 | $72.79 \pm 0.50$ | $72.85 \pm 0.40$ | $71.86 \pm 0.58$ | $69.47 \pm 0.32$ | $70.07 \pm 1.44$ | $71.20 \pm 0.40$ | $\mathbf{73.38 \pm 0.53}$ | $\underline{73.37 \pm 0.65}$ |
| Tiny-ImageNet | $\underline{62.09 \pm 0.45}$ | $61.31 \pm 2.66$ | $60.29 \pm 0.91$ | $-$ | $-$ | $60.39 \pm 0.98$ | $\mathbf{63.47 \pm 0.49}$ | $61.05 \pm 1.25$ |
| SVHN | $96.56 \pm 0.17$ | $96.69 \pm 0.17$ | $96.34 \pm 0.68$ | $96.46 \pm 0.10$ | $95.53 \pm 0.25$ | $96.17 \pm 0.13$ | $\mathbf{96.71 \pm 0.16}$ | $96.69 \pm 0.04$ |
| Mendeley | $75.26 \pm 1.75$ | $73.14 \pm 1.54$ | $75.06 \pm 2.02$ | $75.59 \pm 2.93$ | $73.73 \pm 1.68$ | $73.96 \pm 1.30$ | $\underline{75.75 \pm 1.97}$ | $\mathbf{75.82 \pm 1.51}$ |

Table 5: Comparison of classification performance of SOTA train-time methods in accuracy (%).

## C.2 Comparison on a Non-image Dataset

We conduct experiments on a non-image dataset (20Newsgroup) and compare our method MACSO with the state-of-the-art train-time calibration method MDCA (Hebbalaguppe et al., 2022) (Table 6). On 20Newsgroup, our proposed train-time calibration auxiliary loss MACSO outforms MDCA on both task accuracy and calibration metrics (SCE and ECE). This suggests that our method is not only applicable on image datasets, but rather, a general train-time calibration method that can be used as a regularzier to enhance calibration and potentially, task performance.

| Dataset | FL+MDCA (Hebbalaguppe et al., 2022) | | | FL+MACSO (ours) | | |
|---|---|---|---|---|---|---|
| | SCE $(10^{-3})\downarrow$ | ECE (%) $\downarrow$ | Acc. $\uparrow$ | SCE $\downarrow$ | ECE $\downarrow$ | Acc. $\uparrow$ |
| 20Newsgroup | $1.51 \pm 0.04$ | $12.28 \pm 0.64$ | $65.60 \pm 0.21$ | $\mathbf{1.41 \pm 0.24}$ | $\mathbf{9.73 \pm 5.18}$ | $\mathbf{66.80 \pm 1.75}$ |

Table 6: Calibration performance of our MACSO compared to MDCA on 20Newsgroups dataset.

## C.3 Comparison with Softening with Label Smoothing (LS)

We also report results of our method when replacing our gradually softened target occurrences with LS-based soft targets with the LS's hyper-parameter value of $\alpha \in \{0.05, 0.1\}$ in Table 7. The gradually softened targets leads to better calibration performance than the LS-based soft targets.

| Softening Method | Loss | SCE$(10^{-3})\downarrow$ | ECE(%)$\downarrow$ | Acc.(%)$\uparrow$ |
|---|---|---|---|---|
| Gradually softened targets | FL+MACSO (Linear Corr.) | $\mathbf{3.88 \pm 0.21}$ | $\mathbf{1.06 \pm 0.22}$ | $93.07 \pm 0.31$ |
| | FL+MACSO (Mean) | $\mathbf{3.99 \pm 0.34}$ | $\mathbf{1.17 \pm 0.24}$ | $93.00 \pm 0.26$ |
| | FL+MACSO (KL) | $\mathbf{3.84 \pm 0.40}$ | $\underline{0.94 \pm 0.17}$ | $93.00 \pm 0.28$ |
| LS-based softened targets $(\alpha = 0.05)$ | FL+MACSO (Linear Corr.) | $\underline{4.33 \pm 0.21}$ | $\underline{1.45 \pm 0.12}$ | $92.90 \pm 0.27$ |
| | FL+MACSO (Mean) | $\underline{4.20 \pm 0.28}$ | $\underline{1.22 \pm 0.22}$ | $92.91 \pm 0.21$ |
| | FL+MACSO (KL) | $\underline{4.58 \pm 2.69}$ | $\mathbf{0.81 \pm 0.13}$ | $93.10 \pm 0.25$ |
| LS-based softened targets $(\alpha = 0.1)$ | FL+MACSO (Linear Corr.) | $4.54 \pm 0.33$ | $1.49 \pm 0.23$ | $93.08 \pm 0.30$ |
| | FL+MACSO (Mean) | $\underline{4.04 \pm 0.39}$ | $\underline{1.19 \pm 0.30}$ | $92.99 \pm 0.30$ |
| | FL+MACSO (KL) | $5.83 \pm 0.29$ | $1.57 \pm 0.27$ | $93.02 \pm 0.41$ |

Table 7: Gradually softened targets v.s. Label Smoothing (LS) based softened targets for CIFAR10 dataset.

## C.4 Comparison with a Post-hoc Calibration Method

We compare our method MACSO with a post-hoc calibration method on the datasets of CIFAR10, CIFAR100 and SVHN. Table 8 provides experimental comparisons with Dirichlet Calibration (DC) (Kull et al., 2019) which is a post-hoc calibration method. As demonstrated in the table, our proposed train-time calibration method MACSO outperforms DC.

| Dataset | NLL+DC (Kull et al., 2019) | | FL+DC (Kull et al., 2019) | | NLL/FL + MACSO (ours) | |
|---|---|---|---|---|---|---|
| | SCE $(10^{-3})\downarrow$ | ECE (%) $\downarrow$ | SCE $(10^{-3})\downarrow$ | ECE (%) $\downarrow$ | SCE $(10^{-3})\downarrow$ | ECE (%) $\downarrow$ |
| CIFAR10 | $4.66 \pm 0.42$ | $\mathbf{1.01 \pm 0.20}$ | $4.87 \pm 0.30$ | $1.08 \pm 0.31$ | $\mathbf{3.88 \pm 0.21}$ | $1.06 \pm 0.22$ |
| CIFAR100 | $2.66 \pm 0.07$ | $7.99 \pm 0.60$ | $2.54 \pm 0.06$ | $6.60 \pm 0.56$ | $\mathbf{1.82 \pm 0.05}$ | $\mathbf{1.43 \pm 0.20}$ |
| SVHN | $3.16 \pm 0.30$ | $1.00 \pm 0.17$ | $3.32 \pm 0.37$ | $1.18 \pm 0.25$ | $\mathbf{2.17 \pm 0.22}$ | $\mathbf{0.53 \pm 0.15}$ |

Table 8: Comparison with Dirichlet Calibration (DC) which is a post-hoc calibration method.

## C.5 Comparison of Calibration Metric Convergence with SOTA

Figures 8a and 8b show that the proposed loss function MACSO is better at optimizing both SCE and ECE than the current SOTA train-time calibration methods of MbLS and MDCA. Moreover, compared to others, it consistently decreases both SCE and ECE throughout the evolution of training. Also, Fig. 8c shows the convergence of our auxiliary loss during training. Figures 8a and 8c indicate that, compared to the others, there is more similarity in the behaviour between SCE and our auxiliary

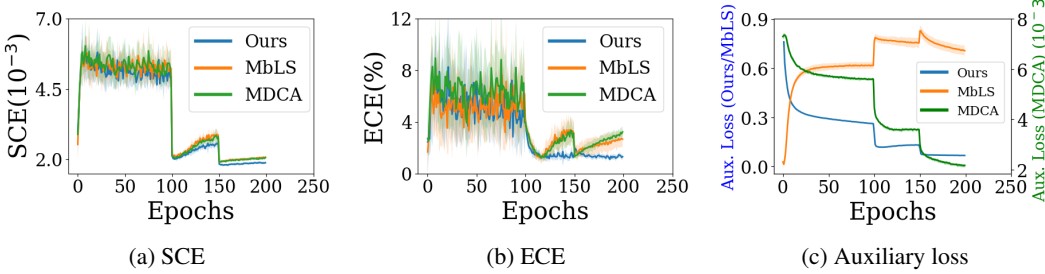

(a) SCE        (b) ECE        (c) Auxiliary loss

Figure 8: Comparison of SCE, ECE and auxiliary loss convergence with SOTA for CIFAR100. Our proposed auxiliary loss MACSO is better at optimizing both ECE and SCE than the SOTA train-time calibration methods (i.e., MbLS and MDCA).

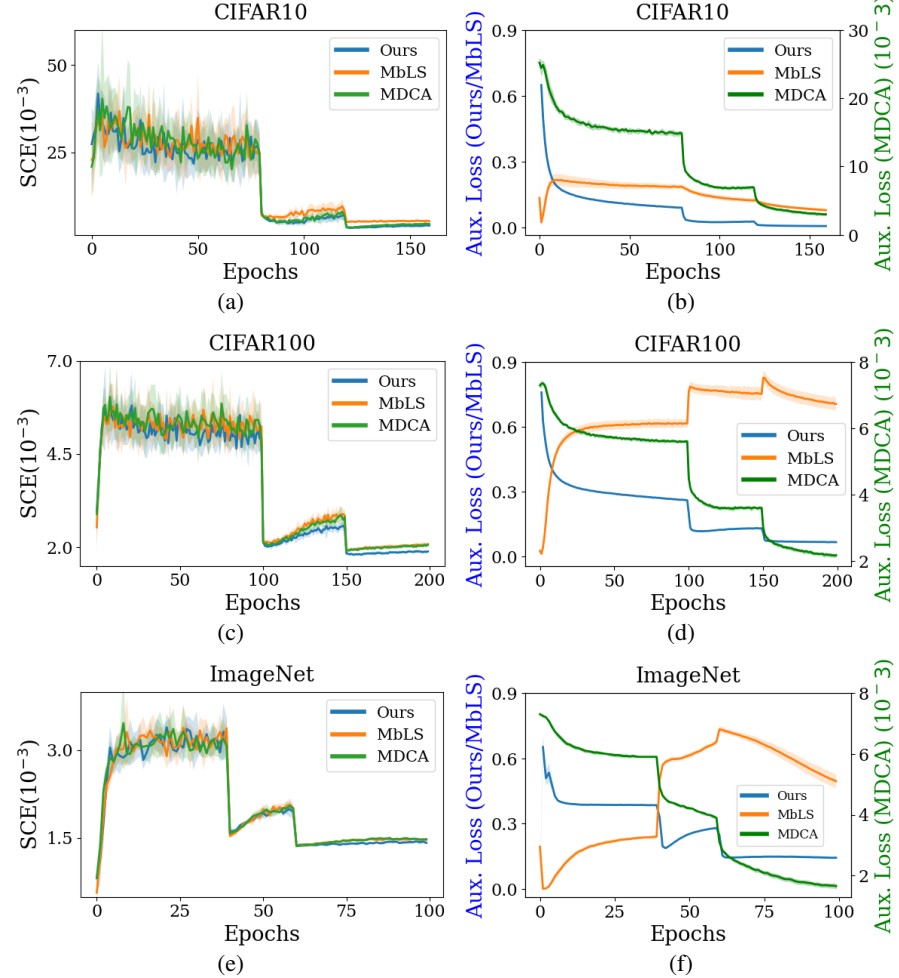

Figure 9: Comparison of SCE and auxiliary loss convergence with SOTA for CIFAR10, CIFAR100 and Tiny-ImageNet. There exists more similarity in the behaviour between SCE and our auxiliary loss MACSO than prior SOTA train-time auxliary calibration losses (MbLS and MDCA), i.e., whenever the SCE values are increasing, the same trend is observed in our auxiliary loss as well.

loss MACSO, i.e., whenever the SCE values are increasing, the same trend could be observed in our auxiliary loss as well. Fig. 9 provides results on more datasets, and it reveals for CIFAR10/100 and Tiny-ImageNet datasets that, compared to the other SOTA auxiliary loss functions, there exists more similarity in the behaviour between SCE and our auxiliary loss MACSO.

# D    ADDITIONAL EXPERIMENTAL RESULTS AND ANALYSIS

## D.1    PARAMETER SENSITIVITY STUDY

We are the first to establish a positive relationship between the conformity of the softened ground truth occurrences distribution and the predicted confidence distribution to the model's calibration error. In our auxiliary loss MACSO, these distributions are computed across the minibatch for each class. In addition, MACSO involves a method-specific hyperparameter $\alpha_{max}$. Henceforth, we conduct a parameter sensitivity study to assess the impact of different batch size and $\alpha_{max}$ values.

**Impact on batch size:** Fig. 10 illustrates the impact of batch size on both calibration performance and discriminative performance. The minibatch size is increased by a factor of 2 from 8 to 1024. We observe the best calibration and accuracy scores with the batch size of 64 or 128. Also, when the minibatch size is 32 or beyond, both accuracy and calibration scores show relatively little variations.

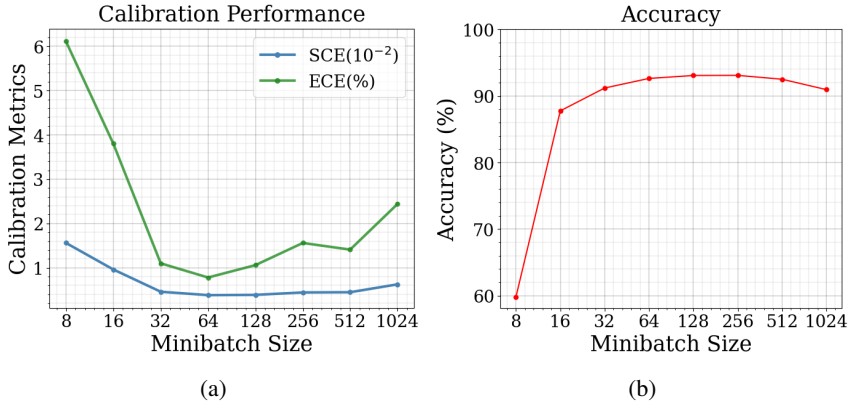

Figure 10: (a) Impact of batch size on calibration performance and (b) the accuracy as a function of minibatch size for FL+MACSO on CIFAR10 using ResNet56.

**Sensitivity on $\alpha_{max}$:** Fig. 11 shows that, our method MACSO is not very sensitive to the choice of $\alpha_{max}$ as varying it does not affect the performance significantly.

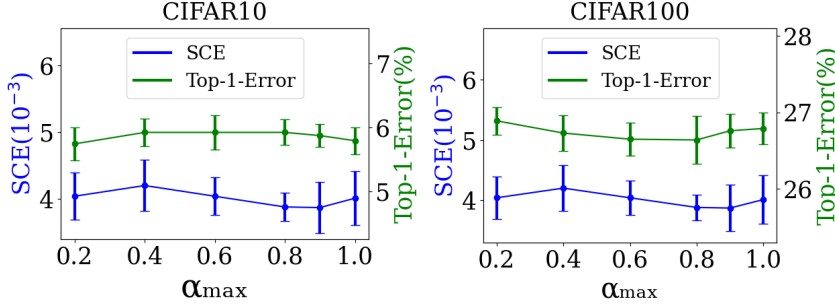

Figure 11: Validation top-1 error and SCE according to $\alpha_{max}$ from 10 trials on CIFAR10/100 for ResNet56.

## D.2    MORE RESULTS ON THE EFFECTIVENESS OF USING GRADUALLY SOFTENED TARGETS

Table 1 (main paper) last row displays results using softened targets for classification loss on CI-FAR10. Table 9 details CIFAR100 and SVHN results, but softened targets for classification loss fail to enhance calibration like our proposed method does. We emphasize that both utilizing gradually softened targets for the calibration term and computing loss with linear correlation lead to our improved calibration results.

| Dataset | NLL$_{soft}$ | | | FL$_{soft}$ | | | FL + MACSO (Ours) | | |
|---|---|---|---|---|---|---|---|---|---|
| | SCE ($10^{-3}$)↓ | ECE (%)↓ | Acc. (%)↑ | SCE ($10^{-3}$)↓ | ECE (%)↓ | Acc. (%)↑ | SCE ($10^{-3}$)↓ | ECE (%)↓ | Acc. (%)↑ |
| CIFAR100 | 1.87 ± 0.05 | 2.37 ± 0.51 | 73.96 ± 0.38 | 4.88 ± 0.08 | 23.98 ± 0.48 | 71.36 ± 0.38 | **1.82 ± 0.05** | **1.43 ± 0.20** | 73.97 ± 0.65 |
| SVHN | 2.97 ± 0.58 | 0.94 ± 0.36 | 96.80 ± 0.07 | 42.80 ± 0.46 | 21.33 ± 0.18 | 96.74 ± 0.08 | **2.17 ± 0.22** | **0.53 ± 0.15** | 96.71± 0.16 |

Table 9: Additional results on CIFAR100 and SVHN, utilizing gradually softened targets for the target-specific classification loss (NLL/FL).

## D.3 ADDITIONAL OUT-OF-DOMAIN RESULTS

We plot the calibration performance as a function of corruption severity level in CIFAR100-C in Fig. 12. Compared to prior SOTA calibration auxiliary losses (i.e., MDCA and MbLS), our method delivers the lowest ECE and SCE scores across all corruption levels.

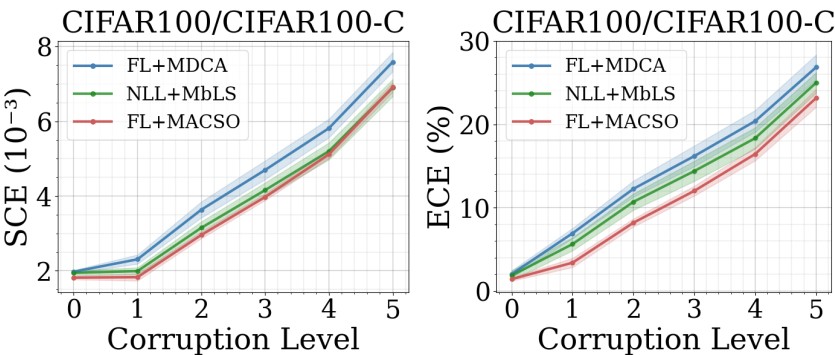

Figure 12: Calibration performance at different image corruption levels in CIFAR100-C dataset. Our method MACSO delivers the lowest ECE and SCE scores across all corruption levels.

## D.4 DETAILED RESULTS ON SVHN

SVHN has a class-imbalance factor of 2.7. The SVHN dataset was used by recent SOTA on train-time calibration, e.g., MDCA Hebbalaguppe et al. (2022), to show calibration performance under class imbalance. We follow prior works to benchmark our method on SVHN. Also, classwise calibration performance (shown in Fig. 3 of the main paper) reflects that our method calibrates all of the classes, even the ones with lower number of samples. In Table. 10, we show the number of samples classwise and the corresponding calibration performance on the SVHN dataset. It can be clearly seen from the '#Training Images' (i.e, the number of training images) column that the dataset is imbalanced and the calibration performance scores show the effectiveness of our method even in the classes with low number of training images.

| Classes | NLL+ MACSO (ours) | FL+ MDCA | NLL+ MDCA | NLL+ MBLS | #Test Images | # Training Images (%) |
|---|---|---|---|---|---|---|
| 0 | **0.13**% | 0.30% | 0.15% | 0.17% | 1744 | 4500 (6.8%) |
| 1 | **0.27**% | 0.79% | 0.31% | 0.29% | 5099 | 12419 (18.84%) |
| 2 | **0.29**% | 0.66% | 0.32% | 0.31% | 4149 | 9591 (14.55%) |
| 3 | 0.29% | 0.70% | 0.39% | **0.28**% | 2882 | 7699 (11.68%) |
| 4 | **0.17**% | 0.40% | 0.19% | 0.22% | 2523 | 6665 (10.11%) |
| 5 | **0.16**% | 0.45% | 0.20% | 0.20% | 2384 | 6187 (9.38%) |
| 6 | 0.22% | 0.45% | **0.19**% | 0.24% | 1977 | 5129 (7.78%) |
| 7 | 0.20% | 0.46% | **0.19**% | 0.25% | 2019 | 5037 (7.64%) |
| 8 | **0.22**% | 0.44% | 0.24% | 0.23% | 1660 | 4536 (6.88%) |
| 9 | **0.21**% | 0.34% | 0.26% | 0.28% | 1595 | 4169 (6.32%) |

Table 10: Class-wise ECE results on SVHN with percentage (%) of training and testing images. It can be clearly seen that the dataset is imbalanced and the calibration performance scores show the effectiveness of our method even in the classes with low number of training images.

## D.5 QUALITATIVE RESULTS FROM MENDELEY (MEDICAL IMAGING)

We show some sample images from Mendeley to show the effectiveness of our proposed method (see Fig. 13). In the left image of Fig. 13, the ground truth is "normal" while incorrect prediction "abnormal" was obtained with both NLL and NLL+MACSO. Confidence estimates of NLL: 0.2034 (normal), 0.7966 (abnormal). Confidence estimates of NLL+MACSO: 0.4888 (normal), 0.5112 (abnormal). In the right image of Fig. 13, the ground truth is "normal", which was predicted by both NLL and NLL+MACSO. Confidence estimates of NLL: 0.8492 (normal), 0.1508 (abnormal). Confidence estimates of NLL+MACSO: 0.9662 (normal), 0.0338 (abnormal). According to the qualitative results, our train-time calibration loss allows the model to have higher confidence values for correct predictions and lower confidence values for incorrect predictions.

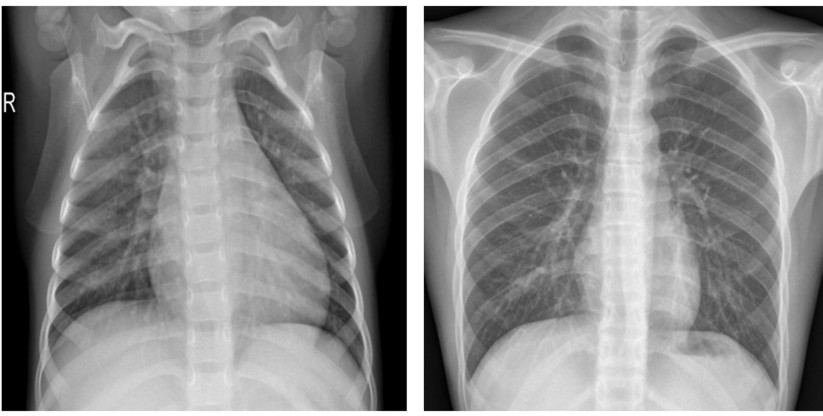

Figure 13: Examples of incorrectly (left) and correctly (right) classified images from Mendeley dataset.

## D.6 MORE RELIABILITY DIAGRAMS AND CONFIDENCE HISTOGRAMS

We plot additional reliability diagrams and confidence histograms in Fig. 14 which compares performance with the current SOTA train-time calibration auxiliary loss functions. Compared to MbLS and MDCA, our method MACSO is more effective in mitigating the overconfidence and further reducing the gap between the overall confidence (dotted line) and accuracy (solid line) of the predictions. Also, in addition to Fig 5 in main manuscript, we plot additional confidence histograms of misclassified predictions in Fig. 15, comparing with the auxiliary loss functions of MbLS and MDCA. Compared to MbLS and MDCA, the confidence values of the incorrect predictions are relatively smaller with our auxiliary loss MACSO.

## D.7 SUPPLEMENTARY RESULTS WITH MORE PRECISION

Table 4 (main manuscript) which compares the individual components of our loss and using KL divergence in our loss, reports the results in one decimal place due to space constraint. Table 11 reports the same comparison with more precision values.

| Dataset | FL+MACSO (Mean) | | | FL+MACSO (Linear Corr.) | | | FL+MACSO (KL) | | |
|---|---|---|---|---|---|---|---|---|---|
| | SCE $(10^{-3})\downarrow$ | ECE (%)$\downarrow$ | Acc. (%)$\uparrow$ | SCE $(10^{-3})\downarrow$ | ECE (%)$\downarrow$ | Acc. (%)$\uparrow$ | SCE $(10^{-3})\downarrow$ | ECE (%)$\downarrow$ | Acc. (%)$\uparrow$ |
| CIFAR10 | $3.99 \pm 0.34$ | $1.17 \pm 0.24$ | $93.00 \pm 0.26$ | $3.88 \pm 0.21$ | $1.06 \pm 0.22$ | $93.07 \pm 0.31$ | $\mathbf{3.84 \pm 0.40}$ | $\mathbf{0.94 \pm 0.17}$ | $93.00 \pm 0.28$ |
| CIFAR100 | $1.95 \pm 0.04$ | $1.89 \pm 0.35$ | $71.90 \pm 0.46$ | $\mathbf{1.82 \pm 0.05}$ | $\mathbf{1.43 \pm 0.20}$ | $73.37 \pm 0.65$ | $1.94 \pm 0.02$ | $1.75 \pm 0.43$ | $72.30 \pm 0.39$ |
| Tiny-ImageNet | $1.48 \pm 0.02$ | $3.18 \pm 0.32$ | $61.09 \pm 0.33$ | $\mathbf{1.44 \pm 0.02}$ | $\mathbf{1.65 \pm 0.27}$ | $61.05 \pm 0.27$ | $1.47 \pm 0.04$ | $2.80 \pm 0.41$ | $61.25 \pm 0.65$ |

Table 11: Impact of different components and linear correlation vs KL divergence in our method (MACSO). We report results with more precision in this table.

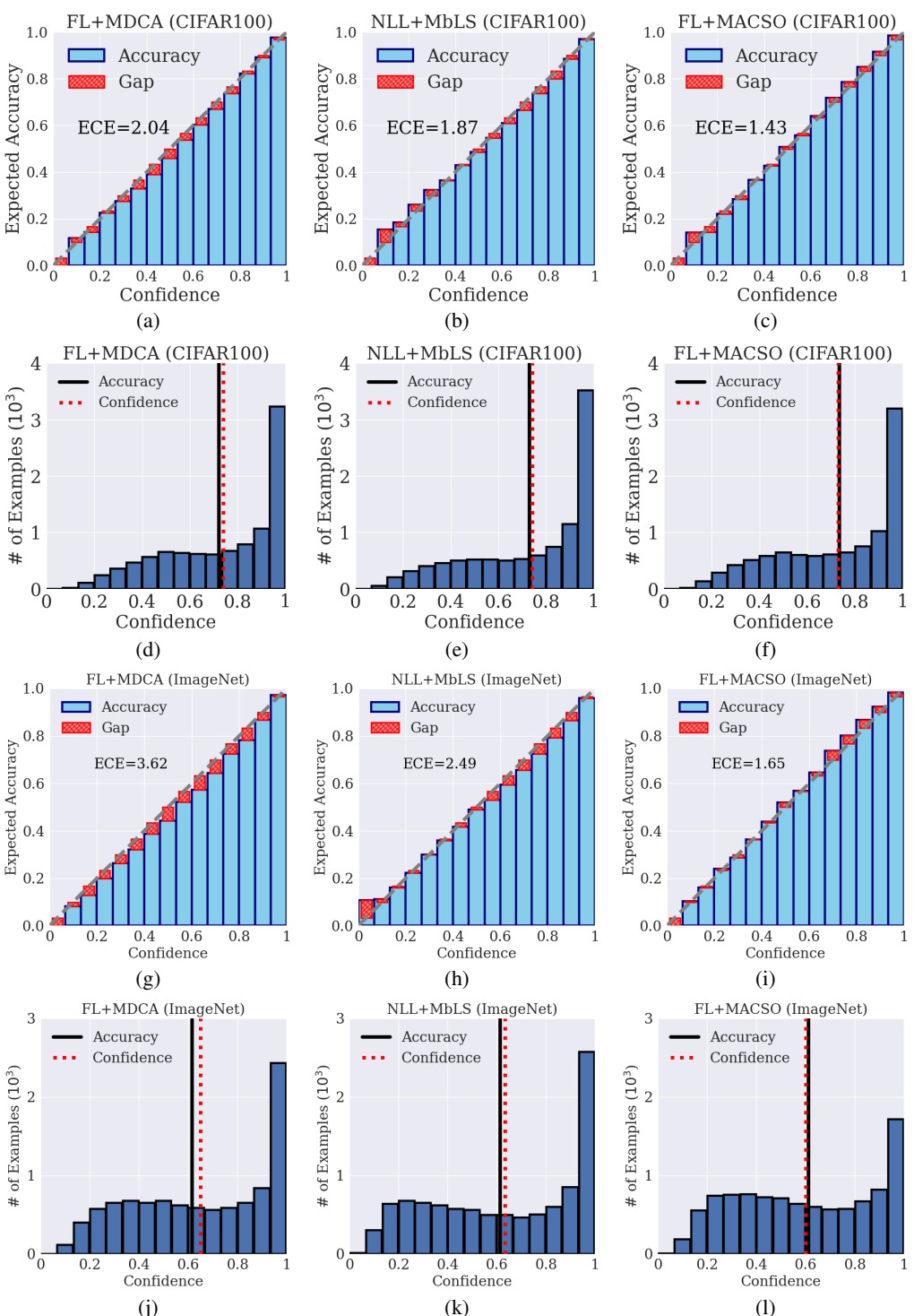

Figure 14: Reliability diagrams (a, b, c, g, h, i) and confidence histograms (d, e, f, j, k, l) of models trained with FL+MDCA, NLL+MbLS and FL+MACSO for CIFAR100 (ResNet56) and Tiny-ImageNet (ResNet50).

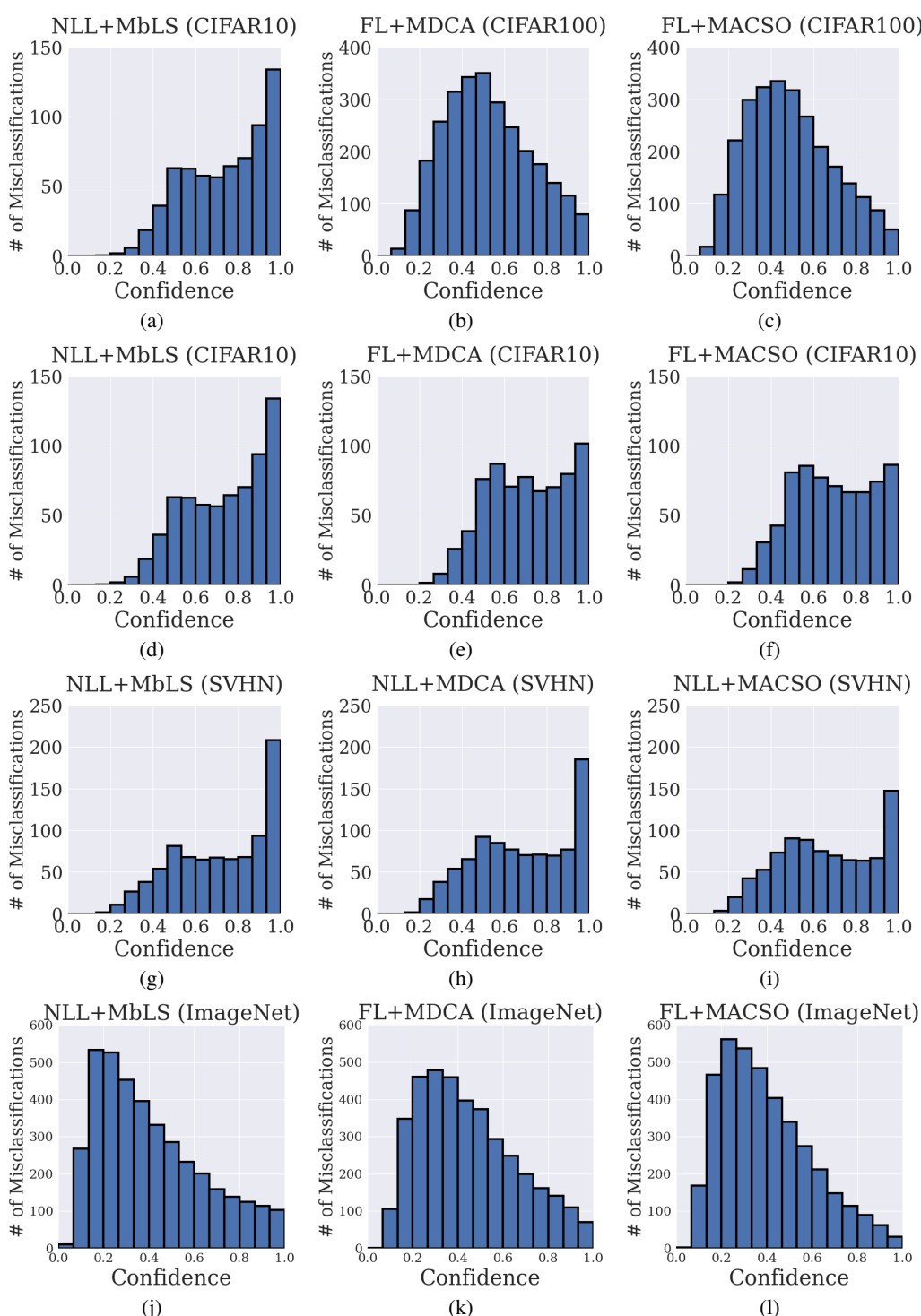

Figure 15: Histogram of confidence values for incorrect predictions in CIFAR100, CIFAR10, SVHN and ImageNet datasets.

## E DISCUSSION ON EXTRA MEMORY REQUIRED IN MACSO

For softening the targets, the softmax predictions from the model at $(t-1)$-th epoch are necessary for training at $t$-th epoch. It can be accomplished using one of the following methods: (1) Keeping the model from the $(t-1)$-th epoch on memory; (2) Saving the $(t-1)$-th predictions on disk.

We have employed the second approach since the datasets of CIFAR10, CIFAR100, Tiny-ImageNet, SVHN and Mendeley V2 are not too large, enabling the ability to store the past predictions on disk. However, for larger benchmarks, the first method avoids heavy disk I/O operations, where additional memory is needed to keep the model in memory at $(t-1)$-th epoch. Both approaches come with their advantages and drawbacks. The first method demands more memory, while the second requires less memory but necessitates additional storage. The decision on which method to use for obtaining past predictions hinges on specific factors, such as the dataset size, in the given scenario.

We compare the memory consumption (in MBs) in Table. 12 on different datasets with FL and FL+MACSO. We observe that MACSO adds very negligible CPU memory overhead. Moreover, we provide time per epoch comparison in Table 13.

| Dataset | FL | FL+MACSO |
|---|---|---|
| CIFAR10 | 15455.99 | 15501.81 (0.30%) |
| CIFAR100 | 16906.62 | 16998.82 (0.55%) |
| Tiny ImageNet | 94702.72 | 94863.48 (0.17%) |

Table 12: Memory consumption (in MBs) overhead by our MACSO over FL.

| Dataset | FL | FL+MDCA | FL+MACSO |
|---|---|---|---|
| CIFAR10 | 16.95s | 18.24s (7.6%) | 18.97s (11.92%) |
| CIFAR100 | 18.56s | 22.5s (21.23%) | 27.6s (48.71%) |
| Tiny ImageNet | 6.34m | 7.34m (15.77%) | 8.56m (35.02%) |

Table 13: Epoch timing in secconds.

# F  DATASET AND IMPLEMENTATION DETAILS

## F.1  DATASETS

**CIFAR10** (Krizhevsky et al., 2009): It comprises $60,000$ images, each having a resolution of $32 \times 32$, equally divided into 10 classes. We use the standard train/validation/test splits of 45K/5K/10K for the experiments. We use the data augmentation techniques of random crops and random horizontal flips on the training set.

**CIFAR100** (Krizhevsky et al., 2009): It also offers $60,000$ images of resolution $32 \times 32$, but equally divided into 100 classes. Standard train/validation/test splits of 45K/5K/10K are used for performing the experiments. Data augmentation techniques such as, random crops and random horizontal flips are applied on the training set.

**Tiny-ImageNet** (Deng et al., 2009): It is a subset of the ImageNet dataset and contains images corresponding to 200 classes with the resolution $64 \times 64$. The training set has $500$ images per class whereas the validation set has $50$ images per class. The provided validation set is used as the test set whereas validation set is generated by randomly sampling $10\%$ of the training set. We employ data augmentation techniques of random crops and random horizontal flips on the training set.

**CIFAR10-C** (Hendrycks & Dietterich, 2019): It is a corrupted version of CIFAR10 test images. For each test image, 18 different corruptions of 5 levels of severity (Hendrycks & Dietterich, 2019) are used to generate its 90 corrupted versions. Thus, the overall dataset contains 900K images.

**CIFAR100-C** (Hendrycks & Dietterich, 2019): It comprises corrupted versions of CIFAR100 test images. For each test image, 18 different corruptions of 5 levels of severity (Hendrycks & Dietterich, 2019) are used to construct its 90 corrupted versions. Thus, the overall dataset contains a total of 900K images.

**Tiny-ImageNet-C** (Hendrycks & Dietterich, 2019): It comprises the corrupted versions of Tiny-ImageNet validation images. 15 different corruptions of 5 levels of severity are used for each image to create its 75 corrupted versions, thus making a total 750K images altogether.

**SVHN** (Netzer et al., 2011): SVHN is a digit classification benchmark comprising $99,289$ images of resolution $32 \times 32$, unequally divided into 10 classes, corresponding to digits from 0 to 9. The provided training and testing sets contain $73,257$ and $26,032$ images, respectively, and $10\%$ of the training set is randomly sampled to be used as the validation set.

**Mendeley V2** (Kermany et al., 2018): This medical dataset comprises optical coherence tomography (OCT) images of the retina and pediatric chest X-ray images. We only use the chest X-ray images with the binary classes (Pneumonia/Normal) in our experiments. The provided training and testing

sets of the chest X-ray images contain 5216 and 624 images respectively, and $10\%$ of the training set is randomly sampled to be used as the validation set.

**20 Newsgroups** (Lang, 1995): 20 Newsgroups is a text classification benchmark comprising $20,000$ news articles, and is equally divided into 20 different news groups. Standard train/validation/test splits of $15,098/900/3,999$ are used for the experiments.

## F.2 Training Details

**Backbone architecture:** We use ResNet backbones for all our experiments. For training Tiny-ImageNet, we use ResNet-50; for Mendeley dataset, ImageNet-pretrained ResNet-50 is used; for training other datasets, we use ResNet-56.

**Training details:** We conduct all experiments involving Tiny-ImageNet dataset on a single V100 GPU, and all the other experiments corresponding to other benchmarks were conducted on a single NVIDIA RTX A6000 GPU. Table 14 summarizes the overall training details where the "scheduler" column indicates the epochs at which the learning rate is decayed and "learning rate decay factor" denotes the factors by which the learning rates are decayed at those corresponding epochs. Whenever SGD optimizer is used (CIFAR10/100, SVHN and Tiny-ImageNet), the momentum is set to $0.9$ and when the Adam optimizer is used (Mendeley), the beta values are set to $0.9$ and $0.999$. We implemented our method on top of MDCA [2] codebase.

| Dataset | Model | Train/Test Batch Size | Optimizer | Epochs | Scheduler | Initial Learning Rate | Learning Rate Decay Factor | Trials |
|---|---|---|---|---|---|---|---|---|
| CIFAR10 | ResNet56 | 128/100 | SGD | 160 | 80 , 120 | 0.1 | 0.1 , 0.1 | 10 |
| CIFAR100 | ResNet56 | 128/100 | SGD | 200 | 100 , 150 | 0.1 | 0.1 , 0.1 | 10 |
| Tiny-ImageNet | ResNet50 | 64/64 | SGD | 100 | 40 , 60 | 0.1 | 0.1 , 0.1 | 5 |
| SVHN | ResNet56 | 128/100 | SGD | 100 | 50 , 70 | 0.1 | 0.1 , 0.1 | 10 |
| Mendeley | ResNet50 | 8/8 | Adam | 20 | 10 | 0.0001 | 0.1 | 10 |

Table 14: Summary of the training details.

**Hyperparameter selection:** For our method, we use $\alpha_{max} = 0.8$ without searching its optimal value on each dataset. We conduct the hyper-parameter search only for method-specific hyper-parameters, i.e., for train-time calibration auxiliary losses, we search $\beta$, which is the loss coefficient of the auxiliary calibration loss term, from the set $\{1, 5, 10, 15, 20 , 25\}$ following the state-of-the-art prior works such as MDCA (Hebbalaguppe et al., 2022). On the CIFAR10 dataset, we did perform a more fine-grained hyper-parameter $\beta$ search. Table 15 shows that a more fine-grained hyper-parameter $\beta$ search is unnecessary. For a fair comparison, we run each experiment 10 trials using different random seeds and report the mean and standard deviation. We choose the hyperparameters with respect to the most accurate models based on the mean validation loss across the 10 trials.

| Dataset | $\beta = 0.2$ | | $\beta = 0.5$ | | $\beta = 0.8$ | | $\beta = 1$ | | $\beta = 5$ | |
|---|---|---|---|---|---|---|---|---|---|---|
| | SCE $(10^{-3})\downarrow$ | ECE $(\%)\downarrow$ | SCE $(10^{-3})\downarrow$ | ECE $(\%)\downarrow$ | SCE $(10^{-3})\downarrow$ | ECE $(\%)\downarrow$ | SCE $(10^{-3})\downarrow$ | ECE $(\%)\downarrow$ | SCE $(10^{-3})\downarrow$ | ECE $(\%)\downarrow$ |
| CIFAR10 | $4.38 \pm 0.34$ | $1.30 \pm 0.31$ | $4.21 \pm 0.39$ | $1.06 \pm 0.21$ | $4.11 \pm 0.40$ | $1.26 \pm 0.28$ | $\mathbf{3.88 \pm 0.21}$ | $\mathbf{1.06 \pm 0.22}$ | $3.88 \pm 0.30$ | $1.09 \pm 0.16$ |
| **Model** | $\beta = 10$ | | $\beta = 15$ | | $\beta = 20$ | | $\beta = 25$ | | | |
| | SCE $(10^{-3})\downarrow$ | ECE $(\%)\downarrow$ | SCE $(10^{-3})\downarrow$ | ECE $(\%)\downarrow$ | SCE $(10^{-3})\downarrow$ | ECE $(\%)\downarrow$ | SCE $(10^{-3})\downarrow$ | ECE $(\%)\downarrow$ | | |
| ResNet56 | $3.93 \pm 0.28$ | $1.20 \pm 0.22$ | $4.33 \pm 0.31$ | $1.49 \pm 0.22$ | $4.21 \pm 0.42$ | $1.26 \pm 0.38$ | $3.93 \pm 0.49$ | $1.32 \pm 0.31$ | | |

Table 15: Calibration performance in SCE $(10^{-3})$ and ECE $(\%)$ for different $\beta$ values. This table shows that a highly detailed hyperparameter search for $\beta$ is unnecessary.

## F.3 Details on Compared Approaches

**Focal Loss** (Lin et al., 2017) and **Label Smoothing** (Müller et al., 2019): The best hyper-parameter value for $\gamma \in \{1, 2, 3\}$ in Focal Loss (FL) and the best hyper-parameter value for $\alpha \in \{0.05, 0.1\}$ in Label Smoothing (LS) is searched using the validation set.

**MMCE** (Kumar et al., 2018): It is used as an auxiliary loss function along with NLL. We choose the weighting/balancing hyper-parameter $\gamma \in \{2, 4\}$ using the validation set.

---

[2]https://github.com/mdca-loss/MDCA-Calibration

**FLSD** (Mukhoti et al., 2020): We set the hyper-parameter of scheduled $\gamma$ in FLSD to 5 for $s_k \in [0, 0.2)$ and 3 for $s_k \in [0.2, 1)$, where $s_k$ represents the confidence score of the correct class.

**MDCA** (Hebbaluppe et al., 2022): Multi-class Difference in Confidence and Accuracy, abbreviated as MDCA is used as an auxiliary loss function, and we use both NLL and FL as the task-specific losses. We choose the weighting/balancing hyper-parameter $\beta \in \{1, 5, 10, 15, 20, 25\}$ and the FL hyper-parameter $\gamma \in \{1, 2, 3\}$ based on the validation loss to report the results. As shown in Table 16, NLL+MDCA provides better results than FL+MDCA for SVHN. Therefore, we choose NLL as the task-specific loss for all the experiments done for SVHN.

| Dataset | NLL+MDCA | | | FL+MDCA | | |
|---|---|---|---|---|---|---|
| | SCE $(10^{-3})\downarrow$ | ECE $(\%)\downarrow$ | Acc. $(\%)\uparrow$ | SCE $(10^{-3})\downarrow$ | ECE $(\%)\downarrow$ | Acc. $(\%)\uparrow$ |
| SVHN | $\mathbf{2.25 \pm 0.69}$ | $\mathbf{0.59 \pm 0.43}$ | $96.53 \pm 0.12$ | $4.99 \pm 1.22$ | $2.16 \pm 0.60$ | $96.48 \pm 0.18$ |

Table 16: Calibration performance of MDCA (Hebbaluppe et al., 2022) when trained as an auxiliary loss with two different task-specific losses, NLL and FL, for the SVHN dataset

**MbLS** (Liu et al., 2022): Margin-based Label Smoothing with the acronym MbLS is used as an auxiliary loss function, and we use NLL as the task-specific loss to report the results. Table 17 reports that NLL+MbLS shows relatively improved calibration performance when compared to FL+MbLS across four different datasets. Therefore, we choose to report results with NLL+MbLS. We select the margin hyper-parameter $m \in \{5, 6, 8, 10, 15, 20, 25\}$ based on validation set and the weight hyper-parameter is fixed at $\lambda = 0.1$ for all experiments (Liu et al., 2022).

| Dataset | NLL+MbLS | | | FL+MbLS | | |
|---|---|---|---|---|---|---|
| | SCE $(10^{-3})\downarrow$ | ECE $(\%)\downarrow$ | Acc. $(\%)\uparrow$ | SCE $(10^{-3})\downarrow$ | ECE $(\%)\downarrow$ | Acc. $(\%)\uparrow$ |
| CIFAR10 | $\mathbf{5.32 \pm 0.08}$ | $\mathbf{1.93 \pm 0.19}$ | $93.39 \pm 0.15$ | $8.30 \pm 0.43$ | $3.01 \pm 0.27$ | $93.33 \pm 0.21$ |
| CIFAR100 | $\mathbf{1.95 \pm 0.06}$ | $\mathbf{1.87 \pm 0.58}$ | $72.94 \pm 0.55$ | $2.16 \pm 0.10$ | $5.06 \pm 0.89$ | $72.08 \pm 0.40$ |
| Tiny-ImageNet | $\mathbf{1.47 \pm 0.03}$ | $\mathbf{2.49 \pm 0.30}$ | $61.22 \pm 1.06$ | $1.51 \pm 0.03$ | $2.52 \pm 0.68$ | $61.35 \pm 1.31$ |
| SVHN | $\mathbf{2.48 \pm 0.38}$ | $\mathbf{0.77 \pm 0.16}$ | $96.75 \pm 0.16$ | $8.22 \pm 0.57$ | $3.86 \pm 0.30$ | $96.59 \pm 0.16$ |

Table 17: Calibration performance of MbLS (Liu et al., 2022) when trained as an auxiliary loss with two different task-specific losses: NLL and FL.

## F.4   MISCELLANEOUS DETAILS

**Correlation plot description:** Fig. 2 (main manuscript) visualizes the relationship (via measuring correlation) between class-wise ECE and (a) MDCA loss (Hebbaluppe et al., 2022), (b) MACSO without linear correlation—where gradually softened target occurrences are used and the mean of distributions are aligned, and (c) MACSO (ours). For each plot, ResNet56 is trained for 160 epochs on CIFAR10 with the respective auxiliary losses along with focal loss as the primary loss. During training, at each epoch, a random minibatch data (128 samples, 10 classes) is saved. Altogether 160 such minibatch data is saved and from each minibatch data (128 samples, 10 classes), class-wise ECE and class-wise auxiliary loss scores are calculated across the minibatch amounting to 10 pairwise values (corresponding to 10 classes) from each epoch. Similarly, for the whole 160 epochs of training, we obtain 1600 pairwise values for class-wise ECE and class-wise auxiliary loss scores which are plotted against one another in Fig. 2 (main manuscript).

**Equations:** For MACSO without linear correlation, the mean of gradually softened target distribution and confidence distribution are aligned, given by Eq. 8:

$$\mathcal{L}_{\text{MACSO w/o Linear Corr.}} = \frac{1}{K} \sum_{j=1}^{K} \left| \frac{1}{N_b} \sum_{i=1}^{N_b} \mathbf{s}_i[j] - \frac{1}{N_b} \sum_{i=1}^{N_b} \tilde{\mathbf{y}}_i[j] \right| \tag{8}$$

For MACSO with KL Divergence, KL divergence is used instead of linear correlation to align the distributions of gradually softened target occurrences and confidences, given by Eq. 9:

$$\mathcal{L}_{\text{MACSO w/ KL}} = \frac{1}{K} \sum_{j=1}^{K} D_{KL}(\tilde{\mathbf{y}}[j] || \mathbf{s}[j]) \qquad (9)$$