# OpenReview forum: "Multiclass Alignment of Confidences and Softened Target Occurrences for Train-time Calibration"
_ICLR.cc/2024/Conference — ICLR 2024 Conference Withdrawn Submission_

### Official Review · Reviewer_fMWK · 2023-10-19

**Soundness:** 2 fair
**Presentation:** 2 fair
**Contribution:** 1 poor
**Rating:** 1
**Confidence:** 5

**Summary:**

This paper present a new train-time calibration method, MACSO, featuring an auxiliary loss formulation that achieves multiclass alignment of confidence distribution and the corresponding distribution of gradually softened target occurrences. However, this method is very similar to PSKD.

**Strengths:**

Prove the effectiveness of the self KD method works in calibration.

**Weaknesses:**

1. The innovation is weak. The method, (Targets softening) is very similar to PSKD [Self-Knowledge Distillation with Progressive Refinement of Targets
]
2. The evaluated architecture is limited, only resent involved.

**Questions:**

See Weaknesses.

---

### Official Review · Reviewer_cBLH · 2023-10-30

**Soundness:** 2 fair
**Presentation:** 3 good
**Contribution:** 2 fair
**Rating:** 3
**Confidence:** 4

**Summary:**

The study introduces a methodology for calibrating a model. The approach is based on a train-time calibration process with an auxiliary loss formulation to achieve multiclass alignment of the confidence distribution and the related distribution of progressively softened target occurrences. The authors perform extensive experiments on various in-domain, class-imbalaced, and out-of-domain scenarios to demonstrate the effectiveness of their proposed method.

**Strengths:**

- Proposed multiclass alignment of the confidence distribution sounds interesting.
- The motivation of the study is clear, and the authors articulate their ideas in a lucid manner.
- The authors conduct comprehensive experiments on a range of scenarios to support their claims.

**Weaknesses:**

- The technical contribution of the paper is limited. The contribution stated in the paper has been published earlier [1] in some way, but no citation is provided for the publication.
- The manuscript has a problem with plagiarism (please see the Ethics concerns for more details).
- This paper mainly focused on the classification task, whereas the previously published paper was based on object detection.

**Questions:**

This paper, in my opinion, duplicates a significant amount of content with an earlier published paper [1]. I was unable to locate any references for the paper, though. Could the authors specifically point out how this study differs from the published one to which they contributed?

[1] B. Pathiraja, M. Gunawardhana and M. Khan, "Multiclass Confidence and Localization Calibration for Object Detection," in 2023 IEEE/CVF Conference on Computer Vision and Pattern Recognition (CVPR), Vancouver, BC, Canada, 2023 pp. 19734-19743.

**Details Of Ethics Concerns:**

I believe that this submission duplicates a substantial amount of material from an earlier published paper [1]. Contributions 1 and 3 are excerpted from contributions 2 and 3 of this paper. Contribution 2 is tangentially related to other purported contributions. Many sections of the manuscript, including the abstract, related work, and preliminary sections, are duplicated or paraphrased from the aforementioned paper. There are also too many other instances of similarity. **Most importantly, the authors did not cite this paper at all.** I believe the author will clarify these issues rigorously.

[1] B. Pathiraja, M. Gunawardhana and M. Khan, "Multiclass Confidence and Localization Calibration for Object Detection," in 2023 IEEE/CVF Conference on Computer Vision and Pattern Recognition (CVPR), Vancouver, BC, Canada, 2023 pp. 19734-19743.

---

### Official Review · Reviewer_MZK5 · 2023-10-31

**Soundness:** 2 fair
**Presentation:** 3 good
**Contribution:** 2 fair
**Rating:** 5
**Confidence:** 4

**Summary:**

This manuscript proposes to use gradually softened ground-truth label to improve the model calibration at train time. The proposed approach also employs Pearson correlation between the distilled soft label and the predictions to replace commonly used KL-divergence. The proposed approach is evaluated on several public toy datasets, and it demonstrates improved calibration performance on some of the scenarios.

**Strengths:**

The task of model calibration under distribution shifts is of significant importance for high-stake predictions.

The proposed approach demonstrates improved calibration performance on public datasets.

Most of the manuscripts are sufficiently readable.

**Weaknesses:**

The proposed work does bear novelty: combining knowledge distillation for label smoothing and replacing NLL/FL with Pearson correlation for smoothed results. However, these contributions may not meet the high standard of ICLR, as both self-distillation and smoothness-oriented losses (e.g., FL) have been widely discussed in uncertainty estimation / calibration, while the authors have not made significant theoretical breakthroughs nor significantly better empirical results in the current shape of submission.

There is unfortunately a lack of clarity in the very core arguments: Sec 3.2: “The loss formulation is inspired by the intuition that as training goes, a model’s prediction becomes refined, and thus the predicted confidence scores can be gradually combined with the ground truth, to form a smoothed target distribution which has an increased entropy compared to the one-hot encoded hard targets, potentially leading to a better calibrated model.” This should be the most important argument supporting the manuscript, but what does this mean? Can the authors please make breaks to improve the clarity?

After Eq. 3: “Additionally, in multiclass calibration, we care about preserving the class relations. Pearson correlation-based loss function allows the model to be guided appropriately to distill those truly informative multiclass relations.” What does “class relation” here mean? The argument of this paragraph is not intuitive. Instead, for multi-class scenarios, KL divergence is also computed for multiple classes as well.

Adding smoothness constraints often comes at the cost of losing sharpness, therefore hurting categorical accuracy: the authors are therefore encouraged to report acc for the upper part of Table 2. Even looking at the lower half of Table 2, the proposed approach sometimes loses sharpness compared with NLL + MbLS. Also, what does NLL/FL in the lower part of Table 2 mean? Are they from NLL or FL or from a linear combination of them?

Despite special considerations for multi-class tasks, when checking SCE (measuring multi-class calibration) and Acc, it is difficult to see if the proposed approach yields better multi-class results than NLL + MbLS

Given that knowledge distillation, including this work, has been widely applied for label smoothing, the authors are encouraged to discuss at least the following similar works, and make comparisons if they find it necessary: [2-4]

[1] ACLS: Adaptive and Conditional Label Smoothing for Network Calibration
[2] Self-Distribution Distillation: Efficient Uncertainty Estimation
[3] Efficient Uncertainty Estimation in Semantic Segmentation via Distillation
[4] Distilling Calibrated Student from an Uncalibrated Teacher

**Questions:**

Given that label smoothing has been employed for long and it can be implemented with a wide range of loss functions, as summarized by [1], what is the major contribution of the proposed approach over existing label smoothing approaches that are discussed in [1]? The authors are encouraged to highlight this in the manuscript.

Given that ECE can be easily abused by degenerative solutions, the authors are encouraged to also report Brier scores which are proper functions for measuring calibration.

Sensitivity on $\alpha$'s should be moved to the main text: this would be a common question raised by most readers.

**Details Of Ethics Concerns:**

The authors are encouraged to discuss the real-world benefit of having better calibrated models in more details.

---

### Official Review · Reviewer_u54s · 2023-11-01

**Soundness:** 2 fair
**Presentation:** 2 fair
**Contribution:** 2 fair
**Rating:** 5
**Confidence:** 3

**Summary:**

The authors propose an auxiliary loss named MACSO for train-time calibration of deep neural networks for image classification. The original label is gradually softened and the multiclass alignment of predictions is calculated to regularize the training process. The method is validated on in and out-of-domain datasets and compared with state-of-the-art models.

**Strengths:**

1. Gradual targets softening during training.
2. Both theoretical and empirical analyses of the advantages of linear correlation over KL divergence.

**Weaknesses:**

1. The introduction contents and design overlap a lot with the cited MDCA work [1].
2. The experiments show marginal gains from the proposed gradual target softening. The benefits of this key component are not strongly demonstrated, weakening the overall contribution.
3. The overall performance is similar to NLL/FL+MDCA, while introducing an additional hyperparameter for the gradual softening. The improvements over existing methods are incremental, and not clearly significant. The authors should provide statistical significance testing and quantify the differences to prior work. Small incremental gains may not be adequately justified as a stand-alone contribution.

[1] Ramya Hebbalaguppe, Jatin Prakash, Neelabh Madan, and Chetan Arora. A stitch in time saves
nine: A train-time regularizing loss for improved neural network calibration. In Proceedings of
the IEEE/CVF Conference on Computer Vision and Pattern Recognition, pp. 16081–16090, 2022.

**Questions:**

The efforts of including the medical image classification task are appreciated as it represents a safety-critical scenario. Why the scale of the SCE and ECE metrics on the medical datasets appear substantially different from the other domains?